# Inverse modeling of the 2021 spring super dust storms in East Asia

Jianbing Jin[1], Mijie Pang[1], Arjo Segers[2], Wei Han[3], Li Fang[1], Baojie Li[1], Haochuan Feng[4], Hai Xiang Lin[5], and Hong Liao[1]

[1]Jiangsu Key Laboratory of Atmospheric Environment Monitoring and Pollution Control, Jiangsu Collaborative Innovation Center of Atmospheric Environment and Equipment Technology, School of Environmental Science and Engineering, Nanjing University of Information Science and Technology, Nanjing, Jiangsu, China
[2]TNO, Department of Climate, Air and Sustainability, The Netherlands
[3]Numerical Weather Prediction Center, Chinese Meteorological Administration, Beijing, China
[4]Department of Sediment Research, China Institute of Water Resources and Hydropower Research, Beijing, China
[5]Delft Institute of Applied Mathematics, Delft University of Technology, Delft, the Netherlands

**Correspondence:** Hong Liao (hongliao@nuist.edu.cn)

**Abstract.** This spring, super dust storms reappeared in East Asia after being absent for one and a half decades. The event caused enormous losses both in Mongolia and in China. Accurate simulation of such super sandstorms is valuable for the quantification of health damages, aviation risks, and profound impacts on the Earth system, but also to reveal the climate driving force and the process of desertification. However, accurate simulation of dust life cycles is challenging mainly due to imperfect knowledge of emissions. In this study, the emissions that lead to the 2021 spring dust storms are estimated through assimilation of MODIS AOD and ground-based $PM_{10}$ concentration data simultaneously. With this, the dust concentrations during these super storms could be reproduced and validated with concentration observations. The multi-observation assimilation is also compared against emission inversion that assimilates AOD or $PM_{10}$ concentration measurements solely, and the added values are analyzed. The emission inversion results reveal that wind blown dust emissions originated from both China and Mongolia during spring 2021. Specifically, 19.9M and 37.5M tons of particles were released in Chinese Gobi and Mongolian Gobi respectively during these severe dust events. By source apportionment it has revealed that Mongolian Gobi posed more severe threats to the densely-populated region Fenwei Plain (FWP) and North China Plain (NCP) located in the northern China than Chinese Gobi. It has estimated that 63% of the dust deposited in FWP was due to transnational transport from Mongolia. For NCP, the long-distance transport dust from Mongolia contributes for about 69% to the dust deposition.

## 1 Introduction

Dust storms occurred as a result of wind erosion liberating dust particles from dry and barren surfaces (Shao et al., 1993). They are relatively common meteorological hazards in arid or semi-arid regions (World Meteorological Organization, 2019). Fine dust particles released from the ground could be lifted several kilometers high, and subsequently carried over long distances, sometimes even across continents (Zhang et al., 2018). The substantial amounts of dust particles, as well as irritating spores, bacteria and viruses carried by the dust storms pose great threats to human health and agriculture (World Meteorological Organization, 2017). Next to these adverse health effects and property losses, the visibility reduction would cause severe

disruption or disorders of the transportation and aviation systems (World Meteorological Organization, 2020). The dust cycle itself is also a key player in the Earth system by influencing the radiative balance (Wu et al., 2016) but also forest and ocean ecosystems (Shao et al., 2011).

Driven by strong cyclones, dust storms form in the Gobi desert could affect East Asia following long distance transport patterns as can be seen in for example Figure 1(a). These events usually occur in spring time, when particles are more erodible under the circumstance of a dry season and sparse vegetation. The frequency and intensity of spring dust storms reached a peak from the 1950s to 1970s, and declined steadily later (Yin et al., 2021). In the past decade, the dust storms were persistently rare, and only 2 strong events occurred in 2015 (Jin et al., 2018) and 2017 (Jin et al., 2019b). The declining trend in dust storm occurrence was co-driven by strict greenness controls in the northern China over the past decades (Shao et al., 2013), climate anomalies (Yin et al., 2021), synoptic disturbances, and other factors not yet explored. However, the ongoing land degradation and desertification in Mongolia (Han et al., 2021) is reported to aggravate the regional dust storms. In spring 2021 however, East Asia experienced an outbreak of severe dust storms after an absence of one and a half decades. Specifically, three super events occurred during March 14 to 16, March 26 to 28 and April 14 to 15, which will be described in Section 2.1 later. The re-occurred spring super dust storms resulted in enormous losses directly. Taking the dust event in March 14-16 for example, 10 people were reported dead and hundred of people reported missing in Mongolia (Chen and Walsh, 2021); the $PM_{10}$ pollutant level in Beijing was brought over 8000 $\mu g/m^3$, and 12 provinces in the northern China were affected with thousands of flights grounded and public transportation systems halted (Jin, 2021). To fully understand the reappeared dust storms is of great interest to health professionals, aviation authorities and policy makers, which not only helps to evaluate property losses, adverse health effects and Earth system impacts, but also to reveal the climate-synoptic driving forces, and finally to build the next generation of the dust early warning system.

Numerical simulation models are commonly used to study dust life cycles. Such models usually consist of an atmospheric transport model that is able to simulate aerosol concentrations, coupled to a dust emission module. The underlying model is often a chemistry-transport model (CTM) which is already used to calculate concentrations of air pollutants, with dust just one of the aerosol species included. Substantial effort has been paid to develop dust models, especially to the dust emission parameterzation which is the most uncertain element in simulating the dust concentrations. Since the early 1990s, several dust emission schemes have been introduced, e.g., MB95 (Marticorena and Bergametti, 1995), Shao96/Shao04 (Shao et al., 1996; Shao, 2004), Ginoux01 (Ginoux et al., 2001) and Zender03 (Zender, 2003). Such emission parameterizations have been included in many global or regional atmospheric transport models, e.g., ECMWF's Integrated Forecast System (IFS) (Morcrette et al., 2008b, a, 2009), BSC-DREAM8b (Pérez et al., 2006; Mona et al., 2014), CUACE/Dust(Zhou et al., 2008; Gong and Zhang, 2008), GEOS-Chem (Fairlie et al., 2007), and LOTOS-EUROS (Timmermans et al., 2017; Manders et al., 2017) as will be used in this study. Due to insufficient knowledge of the actual dust emission and transport, and due to limitations to computing resources to resolve finest-scale variabilities, huge difference might occur when simulations are compared to observations (Niu et al., 2008; Huneeus et al., 2011).

Recent advances in sensor technologies and the continuously decreasing cost of electronic devices have made large scale measurements of dust feasible. For dust storms over East Asia, important observation includes ground-based aerosol optical

properties from the AErosol RObotic NETwork (AERONET) (Dubovik et al., 2000) and Sun-sky radiometer Observation NET-work (SONET) (Li et al., 2018), $PM_{10}$ concentration from the China Ministry of Environmental Protection (MEP) air quality monitoring network (Jin et al., 2018), satellite remote sensing data from polar-orbiting instruments (e.g., Moderate Resolution Imaging Spectroradiometer (MODIS) (Remer et al., 2005) and Cloud-Aerosol Lidar and Infrared Pathfinder Satellite Obser-

vations (CALIPSO) (Winker et al., 2007)), and from geostationary platforms (e.g., Himawari-8 (Bessho et al., 2016) and FY-4 AOPs (Min et al., 2017)). These various types of observations have been used next to the simulation models to analyze the dust storms (Ginoux et al., 2012; Gkikas et al., 2021). Despite their significant roles in characterizing the atmospheric dust load, using only the measurements is not sufficient to obtain a complete four-dimensional insight in the dust plumes, because either the measurements do not cover all area (surface network), only observe once or twice a day (polar orbiting satellites), or

observe only vertically integrated quantities (AOD).

Instead of studying dust storms with either simulation models or observations only, it is useful to combine the measurements and the simulations through data assimilation (Kalnay, 2002). For the purpose of dust storm simulations, the measurements could be used to decrease the uncertainty in the emissions such that the optimized simulation is in better agreement with those measurements (Gong and Zhang, 2008; Lin et al., 2008). For instance, aerosol products from the MODIS instrument onboard

the polar orbiting satellite Terra and Aqua have been widely applied in global or East Asia dust storm assimilation (Di Tomaso et al., 2017; Yumimoto and Takemura, 2015). The geostationary Himawari-8 data has recently also gained popularity and has been used in dust storm detection and assimilation (Yumimoto et al., 2016). In our previous studies described in Jin et al. (2018, 2019a), the ground-based $PM_{10}$ concentration data was assimilated, to estimate the dust source emission for a sandstorm in 2015 over East Asia. Himawari-8 AODs were also assimilated to nudge the dust emission, and the necessity of removing those

AODs in cloudy environment is emphasised in Jin et al. (2019b). An adjoint model was applied to construct a background covariance that was better able to describe the potential source regions for dust emission (Jin et al., 2020). Meanwhile, an imaging morphing based assimilation method was designed which effectively corrected the position error caused in the long-distance dust plume transport (Jin et al., 2021).

While various kinds of dust or aerosol measurements were used in these assimilation experiments, most of these assimilated

only one type of observation at a time. For the sandstorms in East Asia, however, any of the aforementioned measurements cannot provides sufficient information to track these short-term and fast-changing dust plumes completely. For a case in point, the China MEP air quality monitoring network has over 1700 sites for $PM_{10}$ concentrations by now, but these are mainly located in the downwind and densely populated regions, which are far away from the source region of dust in East Asia (the Gobi deserts that can be seen in Figure 1(a)). The network therefore only measure the dust plume when it is already

transported to the downwind urban regions; hence, it is of limited help to characterize the plume near the source regions in rural areas. Besides, the ground-based $PM_{10}$ data only represents the surface dust concentrations and lacks information on the vertical structure. AERONET and SONET instruments provide some information on the vertical structure via column integrated observations, but the network is much sparser than the $PM_{10}$ network, and most of the sites are also far away from the source regions too. The MODIS satellite products provide a global coverage, but again only information on the total column

and therefore no estimate of the plume height or thickness. These polar orbiting instruments also have a limited temporal

coverage; for example, the MODIS Aqua and Terra platforms pass by only around 10:30 and 13:30 (local time). Designed with the wide observing coverage and high temporal resolution, geostationary measuring instruments provide valuable information. However, large uncertainties were found in the Himawari-8 product due to uncertainty in assumptions on aerosol models and surface reflectance estimation in the retrieval algorithm (Zhang et al., 2019). This was also found in Jin et al. (2019b): a strict

observation selection was necessary when assimilating the Himawari-8 AOD values.

Another challenge for dust assimilation is the proper definition of the observations. In general, the commonly used data assimilation schemes all rely on the basic assumption of an unbiased observation. However, all the aforementioned observations, e.g., AOD and $PM_{10}$ concentration, do not only measure dust aerosols, but rather actually the sum of the dust and other fine particles. These originate from for example anthropogenic activities, e.g., industry, vehicles, and households, and from natural

sources such as wildfires and sea spray. These particles are referred to the non-dust fraction in the total aerosols in our study. In the presence of non-dust bias, it is impossible to attribute the difference between the *a priori* simulation and an observation to either this bias or to model deficiencies. The non-dust bias might lead to assimilation that diverge from reality (Lorente-Plazas and Hacker, 2017). However, aerosol measurements are usually directly assimilated, and little progress has been made in bias correction of fully aerosol measurements for their use in dust storm assimilation. Lin et al. (2008) selected only $PM_{10}$

observations for assimilation when at least one occurrence of dust clouds was reported by the local stations. Both machine learning tools and chemical transport models were used for modeling the dynamic non-dust aerosol levels in the $PM_{10}$ concentration measurements, the bias-corrected observations subsequently resulted in more promising assimilation analysis in Jin et al. (2019a). Some efforts have been made to exclude those 'polluted' AODs induced by cloud scenes in the dust assimilation (Jin et al., 2019b); nevertheless, the issue of a non-dust AOD bias remains to large extent unresolved.

To analyze the outbreak of super dust storms in 2021 spring, an emission inversion will be performed through assimilation of multiple observation types. Both the MODIS AOD and the ground-based $PM_{10}$ concentration observations will be used, each providing a different view on the dust plumes. To use the AODs for representing the dust load, an data quality control is designed, which consists of an Ångström-based data screening and a non-dust AOD bias correction. The former is capable of selecting those coarse-mode dust AOD values, and excludes the pixels dominated by fine-mode aerosol from non-dust sources,

while the latter focuses on removing the non-dust baseline in the selected AOD. Similar for $PM_{10}$, a non-dust bias correction as used in Jin et al. (2019a, 2021) is adopted in this work. Based on the posterior emission field obtained in the inversion, the spatial pattern of the emission analyzed for the three events is studied, and the active source regions are identified. In addition, source apportionment simulations are performed, in which the contribution of dust emissions from either Mongolia or China to the dust deposition is calculated for two mega-city clusters, the North China Plain (NCP) and the Fenwei Plain (FWP).

The paper is organized as follows. Section 2 introduces the three super dust events that occurred in this spring, as well as the available dust observations from MODIS and ground-based $PM_{10}$ networks measuring the dust loading from different perspectives. Before AOD is assimilated, quality control combining an Ångström-based screen and a non-dust bias correction are designed to ensure that the observations are representative for the dust load. A brief description of our dust simulation model (LOTOS-EUROS/dust) is presented as well. Section 3 reviews the 4DVar data assimilation based emission inversion system.

In Section 4, the posterior emission field and the estimated AOD and surface dust concentration simulation are evaluated and

**Table 1.** Descriptions of the three severe dust storm events that occurred in China in Spring 2021. Timezone is China Standard Time (CST))

| dust event | affected regions | highest PM$_{10}$ [$\mu$g/m$^3$] | assimilation window | source apportionment simulation timeline |
|:---:|:---:|:---:|:---:|:---:|
| SD1 | NCP | 9993 | March 13, 00:00 to March 15, 23:00 | March 13 to 17 |
| SD2 | NCP, FWP | 9985 | March 26, 00:00 to March 28, 23:00 | March 26 to 29 |
| SD3 | NCP, FWP | 4113 | April 14, 00:00 to April 15, 23:00 | April 14 to 16 |

discussed. Posterior obtained through the multi-observation assimilation is compared against the result from emission inversion that only assimilates PM$_{10}$ or AOD data, and the added values are illustrated. A source apportionment study is performed to obtain information about dust pollution sources and the amount in which they contribute to ambient air pollution in the northern China. Focusing on NCP and FWP, their dominant dust source are identified here.

## 2 Dust events, measurements, and model

### 2.1 Dust events of Spring 2021

The Fenwei Plain (FWP) and the North Chinese Plain (NCP) experienced the most dust affection in China since they face the dust source regions as shown in Figure 1(a). Figure 1(b) shows the recorded PM$_{10}$ concentration average during spring (March-May) in NCP and FWP over 2019-2021. In spite of ongoing air pollutant reduction measures, the PM$_{10}$ levels in NCP and FWP both reached their highest average values of 123 and 119 $\mu$g/m$^3$ in 2021. These mean values are 34% and 27% higher than the means of the previous two years. The reason for this is that East Asia suffered an outbreak of dust storms in the 2021 Spring season, including three large-scale and severe ones. These dust events swept across the northern China, as can be seen from the time series of the hourly PM$_{10}$ concentrations reported by the monitoring stations in NCP and FWP in Figure 1(c)-(d). The timeline in Table 1 shows that these three dust events occurred around March 15, March 28 and April 15, and each of them lasted 3 to 4 days. The highest PM$_{10}$ concentrations reported by the ground-based air quality monitoring network reached values of 9993, 9985, and 4113 $\mu$g/m$^3$. In this study, the three events are referred to as SD1, SD2 and SD3, respectively.

The three dust storms will be studied by combining model simulations and AOD/PM$_{10}$ concentration observations using assimilation, in order to identify the emission sources that affects the densely populated regions in northern China.

### 2.2 Ground-based PM$_{10}$ observations

A huge number of ground-based stations measuring air quality indicators have been established by the China Ministry of Environmental Protection (MEP) since 2013. At present, the monitoring network has grown up to 1800 monitoring sites covering China, of which a part is shown in Figure 1(a). The PM$_{10}$ concentrations observed by the network hence provide valuable information on the dust storms. Snapshots of the PM$_{10}$ measurements for SD1, SD2 and SD3 can be found in Figure

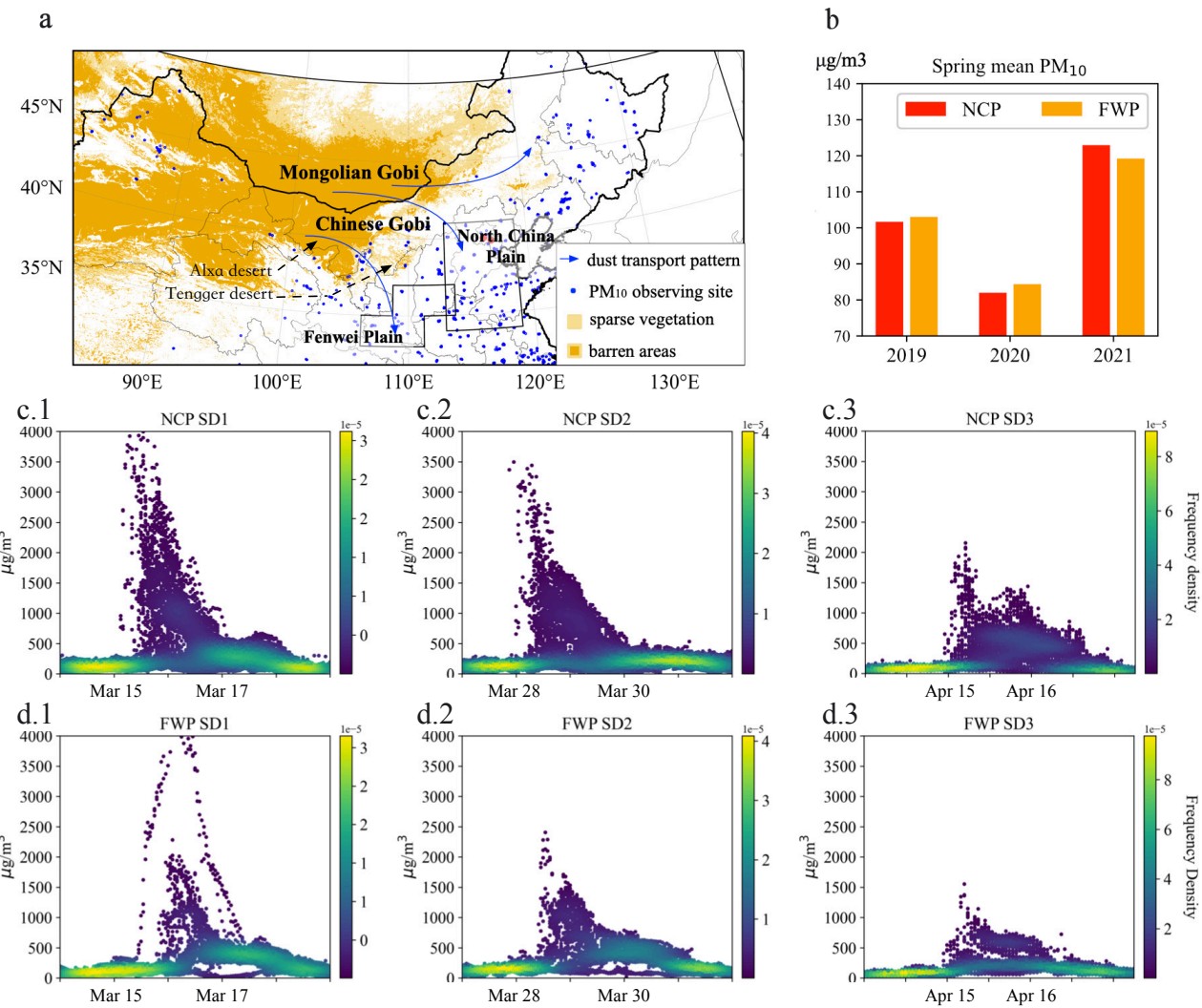

**Figure 1.** (a) Distribution of the potential dust emission source (barren and sparse vegetation landcover) over East Asia and the China MEP observing network over northern China; (b) The spring (March-May) mean $PM_{10}$ concentration observations over NCP and FWP from 2019 to 2021; Time series of the hourly $PM_{10}$ concentration measurements reported by stations in NCP (c) and FWP region (d) during SD1 (column 1), SD2 (column 2) and SD3 (column 3).

2(a.1), Figure 3(a.1) and Figure 4(a.1), respectively. In the observations it is clearly visible where the dust plume is located and how it moves through the regions.

It should be kept in mind that these $PM_{10}$ measurements are actually a sum of dust and other airborne particles like black carbon, sulfate, etc. The $PM_{10}$ data therefore cannot be used directly to represent the dust load. In a previous study (Jin et al., 2019a) it has been shown that removing the non-dust baseline from $PM_{10}$ observations will lead to a more accurate assimilation result, especially in case that dust aerosol is not dominant. In this study, the same observational bias correction is performed to make the $PM_{10}$ measurements fully representative of the dust loads.

The $PM_{10}$ bias correction takes two steps. First, non-dust aerosol levels are calculated using a model simulation with a configuration for air quality simulation (as will be described in Section 2.4), but with the dust tracers disabled. Second, using these simulations, bias-corrected dust observations were calculated by subtracting the non-dust loads from the original $PM_{10}$ concentration observations.

The non-dust aerosol surface concentrations from the simulations, and the dust-only bias-corrected observations of $PM_{10}$ accompanying the original $PM_{10}$ scenes shown in Figure 2(a.1), Figure 3(a.1) and Figure 4(a.1) are available in Supplementary Figure S1, Figure S3 and Figure S5. As example, as shown in Figure S1(a-b), during the SD1 event the non-dust aerosols were also carried southward and ahead of the dust plume due to the strong winds, and the bias-corrected $PM_{10}$ data shows the shape of the dust plume. The shape of the simulated *a priori* dust plume at March 15 10:00 shown in Figure 2(a.2) matches with the observed shape, although the dust concentrations are sometimes very different; this will be discussed in more detail in Section 2.4. Similar features are seen in Figure S3 and Figure S5 for the other events as well.

### 2.3 MODIS AOD observations

### 2.3.1 MODIS Deep Blue AOD product

The Moderate Resolution Imaging Spectroradiometer (MODIS) satellite instruments (Justice et al., 1998) on board of the polar orbiting satellites Terra and Aqua measure atmosphere reflectance at several wavelengths in the range from visible to the near-infrared. The aerosol properties that are retrieved from the MODIS observations have provided high-quality data since 2000 (Terra) and 2002 (Aqua). Designed with a wide swath ($\approx$2330km), MODIS provides near-global observations almost on a daily basis. In this study, the Deep Blue dataset, in the newest MODIS Collection 6.1 operational Level-2 aerosol product is used. The Deep Blue dataset is generated by the Enhanced Deep Blue algorithm (Hsu et al., 2013; Sayer et al., 2014), reporting retrievals over all cloud-free and snow-free land surfaces at a resolution of 10 km. The MODIS Deep Blue dataset is widely used in dust source identification (Ginoux et al., 2012), dust model calibration (Zhang et al., 2018), and emission inversion (Yumimoto and Takemura, 2015; Di Tomaso et al., 2017) and dust field reanalysis (Di Tomaso et al., 2021) through data assimilation.

Snapshots of the MODIS Deep Blue AOD for the SD1, SD2 and SD3 could be found in Figure 2(b.1-c.1), Figure 3(b.1) and Figure 4(b.1). For each of the three events, MODIS AOD provides valuable information for a large part of the domain of interest. For instance, the SD2 plume position and intensity at March 28, 11:00 are clearly identified in Figure 3(b.1), while

the SD3 plume at April 15, 11:00 is observed clearly in Figure 4(b.1). The two AOD scans in Figure 2(b.1-c.1) together also provides the general spatial pattern of the dust plume.

Compared to the ground-based $PM_{10}$ measurements, satellite instruments are designed with a larger observing coverage. However, they also have higher uncertainties in representing the aerosol/dust load (Jin et al., 2019b). In most of the dust model
evaluation/calibration using the satellite data, AOD measurements are directly used for comparing to simulated dust AODs and analyzing the dust strengths. For those regions affected by the severe dust storm, it would be reliable to approximate the dust AODs using the AOD measurements since the amounts of non-dust aerosols are negligible compared to the dust loading; for those areas where dust is not the dominant aerosol, it is then necessary to perform the dust/non-dust AOD discrimination.

### 2.3.2   AOD quality control

While aerosol originating from biomass-burning, urban pollution, and biogenic sources consist mainly of fine-mode particles (with a radius less than 1 $\mu$m), the aerosols in a dust plume are mainly in the coarse modes (particle radius larger than 1 μm) (Dubovik et al., 2002; Jin et al., 2019a). The dominance of one mode over the other can be measured with the Ångström wavelength exponent $\alpha$ (Schuster et al., 2006; Saide et al., 2013; Liu et al., 2019), which describes how the optical thickness depends on the wavelength of the incident light. Ångström exponent is at a range from -0.5 to 2.5, and inversely related to the
average size of the measured aerosol: the smaller the particles, the larger the exponent. Eck et al. (1999) used $\alpha$=0.5 as the threshold for an aerosol mixture dominated by dust, while Schepanski et al. (2007) used $\alpha$=0.6 to detect the presence of dust particles. Ginoux et al. (2012) used $\alpha$=0 to select areas dominated by dust in a single-modal distribution of coarse particles.

Figure 5 shows the four snapshots of retrieved Ångström exponents corresponding to the of AOD values of Figure 2(b.1-c.1), Figure 3(b.1) and Figure 4(b.1). In general, the locations of low Ångström exponent values correspond with the high AOD
values the dust plumes. For example for the snapshot in Figure 5(d), it indicates that a plume of coarse aerosols stays in the Inner Mongolia provinces of China, while fine aerosols are more dominant in the southern regions. This matches with the dust simulation that are shown in Figure 4(b.2). This gives confidence in using the Ångström exponent values for the discrimination between dust and non-dust AOD.

To be able to use the AOD observations in an assimilation focusing on dust only, an Ångström-based data screening and a
non-dust AOD bias correction has been developed and applied. The screening and bias-correction procedures are performed after each other. First the Ångström-based screening selects the pixels with $\alpha < 0.5$ assuming that these are the ones that are dominated by (coarse mode) dust. For these pixels a bias-corrected AOD is calculated by subtracting a non-dust AOD fraction from the selected AODs. Similar to the non-dust $PM_{10}$ simulation, these non-dust AOD baselines are also calculated using a full-chemistry LOTOS-EUROS simulation with the dust tracers disabled. Finally, to make the AOD data resolution
consistent with the model, the MODIS Deep Blue AODs are coarsened by taking the average over the $0.25 \times 0.25$ model grid cell. Snapshots of the non-dust AOD simulation and the bias-corrected AOD measurements for assimilation can be found in Supplementary Figures S2, Figure S4, and Figure S6.

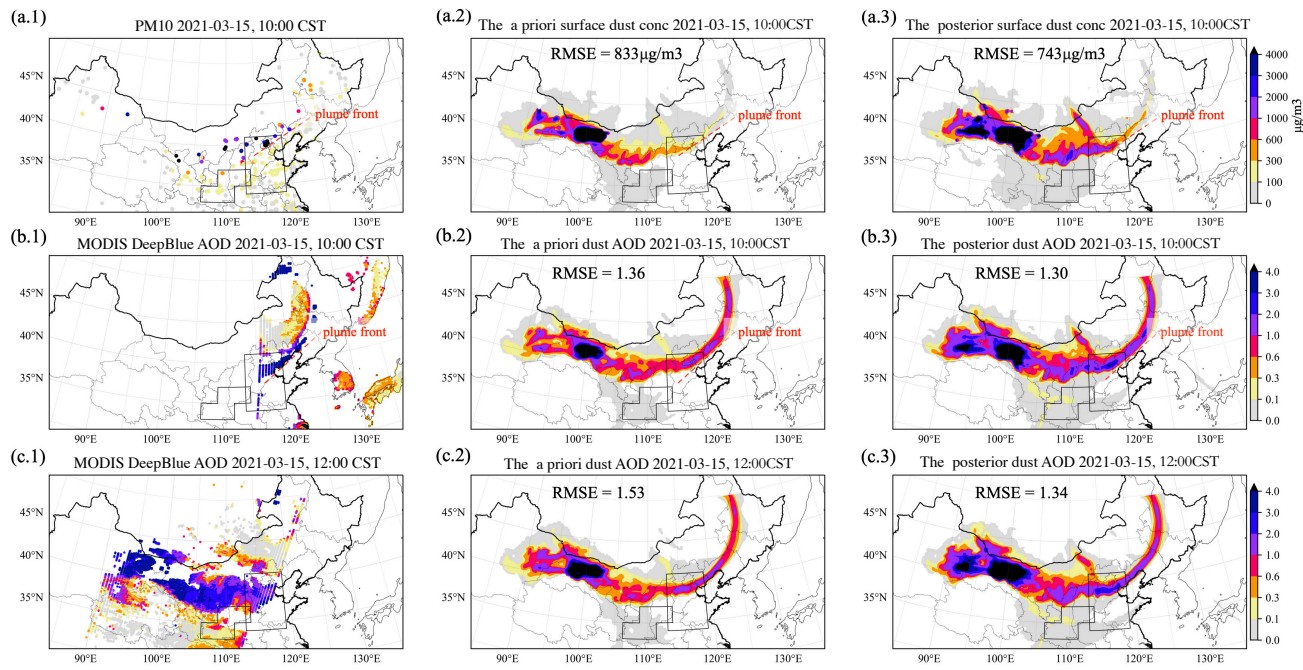

**Figure 2.** Snapshots of observations and simulations during severe dust event 1 (SD1) at 2021 March 15. Row (a) shows $PM_{10}$ concentrations observed by ground network at 10:00 CST (column 1), and *a priori* (column 2) and *posterior* (column 3) simulations by LOTOS-EUROS/dust model. Similar, row (b) shows for the same hour AOD at 550nm from MODIS Deep Blue observations (column 1), and simulations (columns 2-3) at 10:00; row (c) shows the same for the MODIS overpass at 12:00.

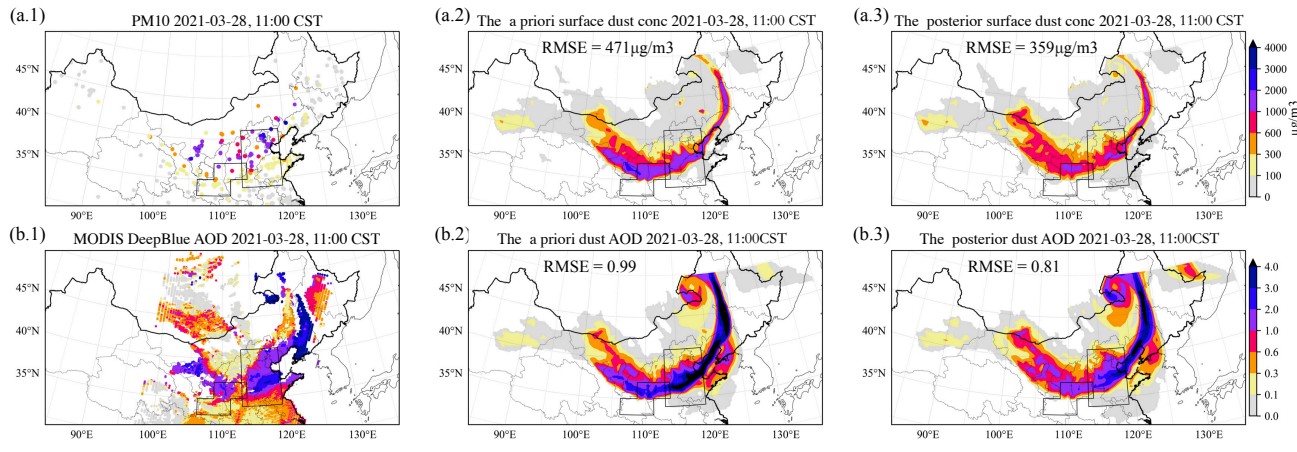

**Figure 3.** Snapshots of observations and simulations during severe dust event 2 (SD2) at 2021 March 28. Row (a) shows $PM_{10}$ concentrations observed by ground network at 11:00 CST (column 1), *a priori* (column 2) and *posterior* (column 3) simulations by LOTOS-EUROS/dust model. Row (b) shows for the same hour AOD at 550nm from MODIS Deep Blue observations (column 1), and simulations (columns 2-3).

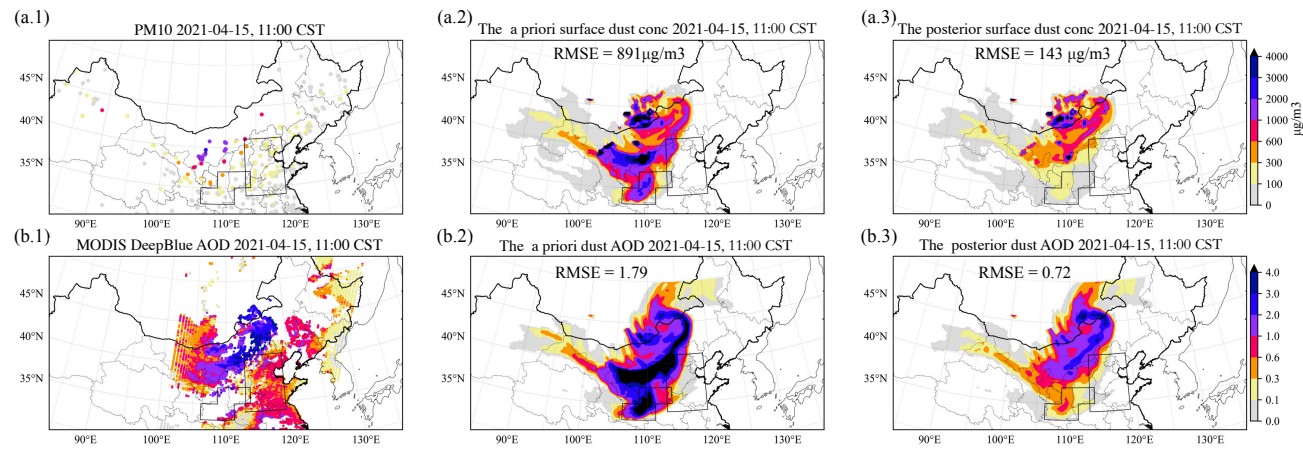

**Figure 4.** Snapshots of observations and simulations during severe dust event 3 (SD3) at 2021 April 15. Row (a) shows PM$_{10}$ concentrations observed by ground network at 11:00 CST (column 1), *a priori* (column 2) and *posterior* (column 3) simulations by LOTOS-EUROS/dust model. Row (b) shows for the same hour AOD at 550nm from MODIS Deep Blue observations (column 1), and simulations (columns 2-3).

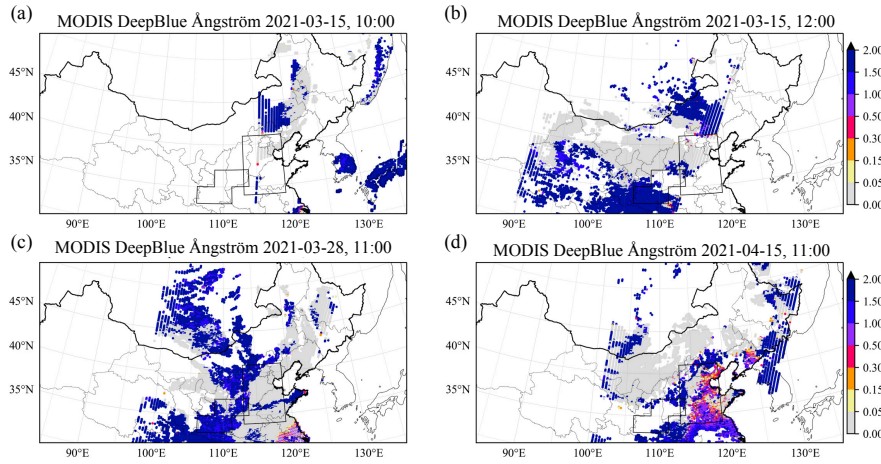

**Figure 5.** Snapshots of MODIS Deep Blue Ångström exponent accompanying AOD scenes shown in Figure 2(a.1-b.1), Figure 3(a.1) and Figure 4(a.1).

## 2.4 Dust simulation model

The regional chemical transport model LOTOS-EUROS v2.1 (Manders et al., 2017) is used to simulate dust concentrations. LOTOS-EUROS has been used for a wide range of applications supporting scientific research and operational air quality forecasts over Europe, China, and other regions. Daily operational forecasts over China used to be released via the MarcoPolo–Panda projects (Timmermans et al., 2017; Brasseur et al., 2019). Additionally, it is also implemented in the World Meteorological Organization (WMO) Sand and Dust Storm Warning Advisory and Assessment System to provide short-time forecasting of the dust load over the area of North Africa, Middle East, and Europe; the online forecast product is delivered via http://sds-was.aemet.es/forecast-products/dust-forecasts (last access: July 2021).

To simulate dust concentrations over East Asia, the model is configured on a domain from 15°N to 50°N and 70 °E to 140 °E, with a resolution of 0.25°×0.25°. Vertically, the model consists of eight layers, with a top at 10 km. Our regional model has zero boundary conditions by assuming all dust aerosols are emitted regionally and the external dust flows can be ignored. Though BSC-DREAM8b simulation, which is released via https://ess.bsc.es/bsc-dust-daily-forecast (last access: Jan 2022), indicates that part of the dust plume released in Middle East is likely to be transported to the western China during SD1, but those particles did not go as far as FWP or NCP. Boundary conditions from a global dust model are in demand if we are focusing on the western China in our future work. The dust simulation is driven by European Centre for Medium-Ranged Weather Forecasts (ECMWF) operational forecasts over 3–12 h, retrieved at a regular longitude–latitude grid with a resolution of about 7 km. An interface to the ECMWF output set is designed, which not only interpolates the default 3h ECMWF short-term forecast meteorology to hourly values but also averages the forecast to fit the LOTOS-EUROS spatial resolutions (Manders et al., 2017). Physical processes included are wind-blown dust emission, diffusion, advection, dry and wet deposition, and sedimentation.

## 2.5 Dust emission and uncertainty

The goal of this work is to calculate the optimal emission field that best fits the *a priori* model simulation and the observation. It is then necessary to define and quantify the uncertainty in the dust simulations. In this study, we define the main model uncertainty to be in the parametrization of the dust emissions. Although other model processes such as transport and deposition are uncertain too, for the events studied here these assumed to be of less importance than the location and the intensity of dust emission.

The physical basis of the dust emission model adopted in LOTOS-EUROS is the parameterization scheme by Zender (2003). The dust flux $f$ is calculated as a function of horizontal saltation $f_h$ (Marticorena and Bergametti, 1995), the sandblasting efficiency $a$, a terrain preference $\mathcal{S}$, and an erodible surface fraction $\mathcal{C}$:

$$f = f_h \cdot a \cdot \mathcal{S} \cdot \mathcal{C} \tag{1}$$

The dust saltation rate $f_h$ is proportional to the third power of the wind friction velocity $u$, as long as this exceeds a certain (surface depended) friction velocity threshold $u_t$:

$$
f_h = \begin{cases} 0 & u \le u_t \\ \frac{p_a}{g}\, u^3 \left(1 + \frac{u_t}{u}\right) \left(1 - \frac{u_t^2}{u^2}\right) & u > u_t \end{cases}
\tag{2}
$$

The friction velocity threshold controls if dust is released from a surface at all, and how strong the emission is. In Jin et al. (2018) it was shown that the uncertainty in the friction velocity threshold parametrization is the main reason that prohibits accurate dust emission forecasts. Although other controlling factors, e.g., the wind field uncertainty, will also introduce uncertainty in the emission partially, these were found to be of less important in the emission error quantification (Jin et al., 2019b).

## 3 Data assimilation algorithms

### 3.1 Assimilation method

The assimilation system that will be used to combine $PM_{10}$ and AOD measurements with dust simulations, is based on the reduced-tangent-linearization four-dimensional variational (4DVar) data assimilation developed in Jin et al. (2018). The goal of the 4DVar technique is to find the maximum likelihood estimation of a state vector, which is here the dust emission field $\boldsymbol{f}$, given both the available AOD and $PM_{10}$ measurements over a time window. The optimal emission $\boldsymbol{f}$ is calculated by minimizing the cost function:

$$
\mathcal{J}(\boldsymbol{f}) = \mathcal{J}_b(\boldsymbol{f}) + \mathcal{J}_o^{\mathrm{PM}}(\boldsymbol{f}) + \mathcal{J}_o^{\mathrm{AOD}}(\boldsymbol{f})
\tag{3}
$$

In here, $\mathcal{J}_b$ represents the background term as follows:

$$
\mathcal{J}_b(\boldsymbol{f}) = \frac{1}{2}(\boldsymbol{f} - \boldsymbol{f}_b)^{\mathrm{T}} \mathbf{B}^{-1} (\boldsymbol{f} - \boldsymbol{f}_b)
\tag{4}
$$

$\boldsymbol{f}_b$ represents the *a priori* or *background* dust emission vector which follows the calculation described in Section 2.4. As in Jin et al. (2018), the errors in dust emission field are assumed to be only caused by the uncertainty in the friction velocity threshold in Eq.2, and it can be compensated by introducing a spatially varying multiplicative factor $\beta$ as:

$$
u_t^{\mathrm{true}}(i,t) = \beta(i) \cdot u_t(i,t)
\tag{5}
$$

where $u_t(i,t)$ denotes the model parameterized friction velocity threshold in a given grid cell $i$ at instant $t$ while $u_t^{\mathrm{true}}(i,t)$ denotes the true value. The $\beta$s values are defined to be random variables with a mean of 1.0 and a standard deviation $\sigma_\beta = 0.1$. This empirical standard deviation was found to provide sufficient variations to resolve the observation-simulation difference. A background covariance matrix $\mathbf{B}_\beta$ is then formulated by combining the constant standard deviation and a correlation matrix $\mathbf{C}$:

$$
\mathbf{B}_\beta(i,j) = \sigma_\beta^2 \cdot \mathbf{C}(i,j)
\tag{6}
$$

in here, $\mathbf{C}(i,j)$ denotes a distance-based spatial correlation between the $\beta$s in two grid cells $i$ and $j$, which is defined as:

$$\mathbf{C}(i,j) = e^{-(d_{i,j}/L)^2/2} \tag{7}$$

here $d_{i,j}$ represents the distance between two grid cells $i$ and $j$. In Jin et al. (2018), the correlation length scale $L$ was configured to be 800 km, which was found to be suitable to simulate the main characteristics of the dust event studied there. In this study, however, a smaller length scale $L = 300$ km is used which gives a higher spatial degree of freedom, while it can still be resolved by the assimilation due to the larger amount of MODIS AOD and PM$_{10}$ concentration measurements.

With the covariance matrix $\mathbf{B}_\beta$, an ensemble ( $N=200$ ) of samples of $\beta$ is generated randomly. These ensemble samples are then applied in our dust model, and each of them produces a emission forecast. The covariance of these emission fields are approximated as follows:

$$\mathbf{B} \approx \frac{1}{N-1} \sum_{i=1}^{N} (\boldsymbol{f}_{u_t,i} - \bar{\boldsymbol{f}}_{u_t})(\boldsymbol{f}_{u_t,i} - \bar{\boldsymbol{f}}_{u_t})^T \tag{8}$$

where $\boldsymbol{f}_{u_t,i}$ represents the emission vector computed using friction velocity threshold ensemble member $i$, while $\bar{\boldsymbol{f}}_{u_t}$ is the ensemble mean.

The observation term of the cost function, $\mathcal{J}_o^{\mathrm{PM}}$ and $\mathcal{J}_o^{\mathrm{AOD}}$ quantify mismatch between dust simulation and PM$_{10}$/AOD measurements:

$$\mathcal{J}_o^{\mathrm{PM}}(\boldsymbol{f}) = \frac{1}{2} \sum_{i=1}^{m} \{ \boldsymbol{y}_i^{\mathrm{PM}} - \mathcal{H}_i^{\mathrm{PM}} \mathcal{M}_i(\boldsymbol{f}) \}^{\mathrm{T}} \mathbf{O}_i^{\mathrm{PM}^{-1}} \{ \boldsymbol{y}_i^{\mathrm{PM}} - \mathcal{H}_i^{\mathrm{PM}} \mathcal{M}_i(\boldsymbol{f}) \} \tag{9}$$

$$\mathcal{J}_o^{\mathrm{AOD}}(\boldsymbol{f}) = \frac{1}{2} \sum_{i=1}^{n} \{ \boldsymbol{y}_i^{\mathrm{AOD}} - \mathcal{H}_i^{\mathrm{AOD}} \mathcal{M}_i(\boldsymbol{f}) \}^{\mathrm{T}} \mathbf{O}_i^{\mathrm{AOD}^{-1}} \{ \boldsymbol{y}_i^{\mathrm{AOD}} - \mathcal{H}_i^{\mathrm{AOD}} \mathcal{M}_i(\boldsymbol{f}) \} \tag{10}$$

where $m$ and $n$ are the number of time steps within the assimilation window; $\boldsymbol{y}^{\mathrm{PM}}$ and $\boldsymbol{y}^{\mathrm{AOD}}$ contain the pre-processed PM$_{10}$ and MODIS AOD measurements; $\mathcal{M}$ denotes the LOTOS-EUROS/dust transport model that is driven by the emission $\boldsymbol{f}$, and $\mathcal{H}^{\mathrm{PM}}$ and $\mathcal{H}^{\mathrm{AOD}}$ are the observation operators that convert simulated dust concentrations into PM$_{10}$ and AOD observation space. The PM$_{10}$ and AOD observation mismatch term are weighted by observation error covariance $\mathbf{O}^{\mathrm{PM}}$ and $\mathbf{O}^{\mathrm{AOD}}$. The uncertainties in the observations are assumed to be independent, and hence both $\mathbf{O}^{\mathrm{PM}}$ and $\mathbf{O}^{\mathrm{AOD}}$ are diagonal matrices.

Both the instrument and representing errors are considered when the observation error covariance, $\mathbf{O}^{\mathrm{PM}}$ and $\mathbf{O}^{\mathrm{AOD}}$, are designed. The uncertainty (square root of the individual diagonal element in $\mathbf{O}^{\mathrm{PM}}$) of the pre-processed PM$_{10}$ measurements for assimilation is assumed to be due to uncertainty in the PM$_{10}$ data and the non-dust PM$_{10}$ bias correction. We have used $\sigma^{\mathrm{PM}} = \max(\ 200,\ 10\% \cdot y^{\mathrm{PM}} + 180\ )$ to characterize the uncertainty of PM$_{10}$ data. It follows the choice of 10% in our previous study (Jin et al., 2018) with uncertainty inflated for this application. This is mainly to prevent the posterior from getting too close to the low-value PM$_{10}$ observations and hence being model divergent. In addition, the uncertainty of the non-dust PM$_{10}$ simulation $\sigma^{\mathrm{BC}}$ that is introduced in Section 2.2 is set to 40% following the aerosol simulation analysis over China using LOTOS-EUROS (Timmermans et al., 2017). The integrated uncertainty $\sigma^{\mathrm{integrated}}$ for using the bias-corrected PM$_{10}$ to represent the dust load is then calculated as:

$$\sigma^{\mathrm{integrated}} = \{\ (\sigma^{\mathrm{PM}})^2 + (\sigma^{\mathrm{BC}})^2\ \}^{0.5} \tag{11}$$

Snapshots of $\sigma^{PM}$ and $\sigma^{integrated}$ distribution accompanying the $PM_{10}$ measurements shown in Figure 2(a.1), Figure 3(a.1) and Figure 4(a.1) in the three dust events are shown in Figure S1(c-d), Figure S3(c-d) and Figure S5(c-d), respectively.

The integrated uncertainty of AOD measurements for assimilation is also calculated as the sum of the instrument error and the error of the non-dust AOD bias correction. The former is taken directly from the MODIS Deep Blue product, while the uncertainty of non-dust AOD simulation is set to 40% as well. Snapshots of the AOD instrument uncertainty and integrated uncertainty with respect to the AOD observations in the three dust events can be found in Figure S2(c-d), Figure S4(c-d) and Figure S6(c-d).

## 3.2 Assimilation window

Table 1 shows the timeline of the three severe dust events studied here. These dust events have a short duration, and therefore a single assimilation window with a length of 72, 72 and 48 hours are used, respectively. Dust emissions occur at the start of the assimilation window when a plume is lifted high to be carried downwind. The $PM_{10}$ network is only able to observe the plume when it has moved downwind, and is already far away from the source region. Therefore, assimilation windows covering both the time of dust emission and the moment of observation are necessary. When the dust plume is carried further southward or eastward, the error of the simulated dust concentration grows steadily due to the accumulation of the dust transport and deposition uncertainty. Therefore, AOD or $PM_{10}$ measurements out of the assimilation window help little in the emission inversion and therefore are not used here.

## 4 Results and discussions

### 4.1 Dust storm inverse modeling

Using the assimilation system introduced in Section 3, the emission inversions have been performed by assimilating the bias-corrected $PM_{10}$ and MODIS AODs processed in Section 2.2 and Section 2.3, and it is referred as *multi-observation* assimilation in this study. First, the posterior emission analysis is carried out in Section 4.1.1, then the dust simulation driven by the posterior emission result are illustrated in Section 4.1.2.

#### 4.1.1 Dust emission analysis

To assess the spatial pattern of the dust emission, the accumulated emissions over the assimilation window are calculated for each of the dust events. Figure 6 shows the map of the a priori and posterior accumulated dust emission in SD1, SD2 and SD3 over the potential source regions. The a priori model simulation indicates dust emission took place over both the Alxa desert (part of Chinese Gobi) and the central of Mongolian Gobi desert during SD1, and their maximum emission flux exceeded 200 g/m$^2$. Through assimilating the MODIS AODs and ground-based $PM_{10}$ concentration measurements, the emission field is estimated. It indicates that emission took place in more grid cells that are located in the Alxa desert and in the eastern Mongolia.

However, the dust plume released from the Alxa desert did not move far towards the south or the east, while Mongolian dust was the main source of dust affection in the northern China as will be discussed in Section 4.3.

For the SD2 event, the accumulated a priori emissions are rather high, especially around the border between China and Mongolia. The emission accumulations in several grid cells here are in the order of 300 g/m$^2$, which results in the overestimation of surface dust concentration and AOD simulation as shown in Figure 3(a.2)-(b.2). In contrast, the posterior simulation estimates that the border region is almost free of dust emission during the event, and the dust plume is actually attributed to the dust emission from the northern Mongolia. The Alxa desert also contributes partially to the dust plume that effected the FWP region.

For the SD3 event, the a priori emission simulation suggests that most of the dust originated from the source regions in China rather than from the bare lands in Mongolia. Especially in Tengger (also part of the Chinese Gobi) and Alxa desert, the accumulated emissions reach values over 200 g/m$^2$. By assimilating the MODIS AOD and PM$_{10}$ concentration data, the posterior dust emission field is updated and the assimilated estimate of the accumulation map is plotted in Figure 6(c.2). What is interesting about the result in the posterior map is that a much smaller amount of dust is estimated to be emitted from Tengger and Alxa desert, while sparsely vegetated regions in the northeastern part of the Inner Mongolia province is estimated to be a significant source. Dust from these regions is estimated to be transported towards NCP and Northeast China, as will be discussed later. Close inspection of the posterior emission map shows that the emission over the Mongolia and China border region is also one of the main source in the SD3 event.

Apart from the spatial patterns of dust emission, the total mass of the emissions from Mongolia and China are calculated and shown in Figure 6(d), which helps to evaluate the emission intensity of dust in these two countries. Although the spatial patterns have been strongly changed by the assimilation of the MODIS and PM$_{10}$ concentration measurements, the posterior emission sums per country and event are in most cases close to the *a priori* values. An except is the value for the China in SD3, for which the total emission is decreased from 15.4M tons to 7.1M tons by the assimilation. The emission sums show that Mongolian Gobi is a stronger source of dust (37.5M ton) than the Chinese Gobi (19.9M ton).

The dust emission inversion successfully optimized the dust simulation using different types of observations of the dust plumes by adjusting the emission fields. The posterior emission field not only helps us exploit the spatial pattern of active dust sources, but also to simulate the long-distance dust transport more accurately. This could be used for evaluation of the threats that dust imposes on the human health, the transport system and the Earth system.

### 4.1.2 Simulated dust field

The impact of the assimilation has been evaluated by comparing the simulated observations from the *a priori* and *posterior* simulations with the (bias corrected) observations. The purpose of this paper is to have a emission field as accurate as possible, therefore, we assimilated all available PM$_{10}$ and MODIS AOD observations instead of leaving a subset of them for independent validation. To quantify the performance, the root mean square error (RMSE) that calculates the deviation of the simulation with respect to the bias-corrected measurements is calculated.

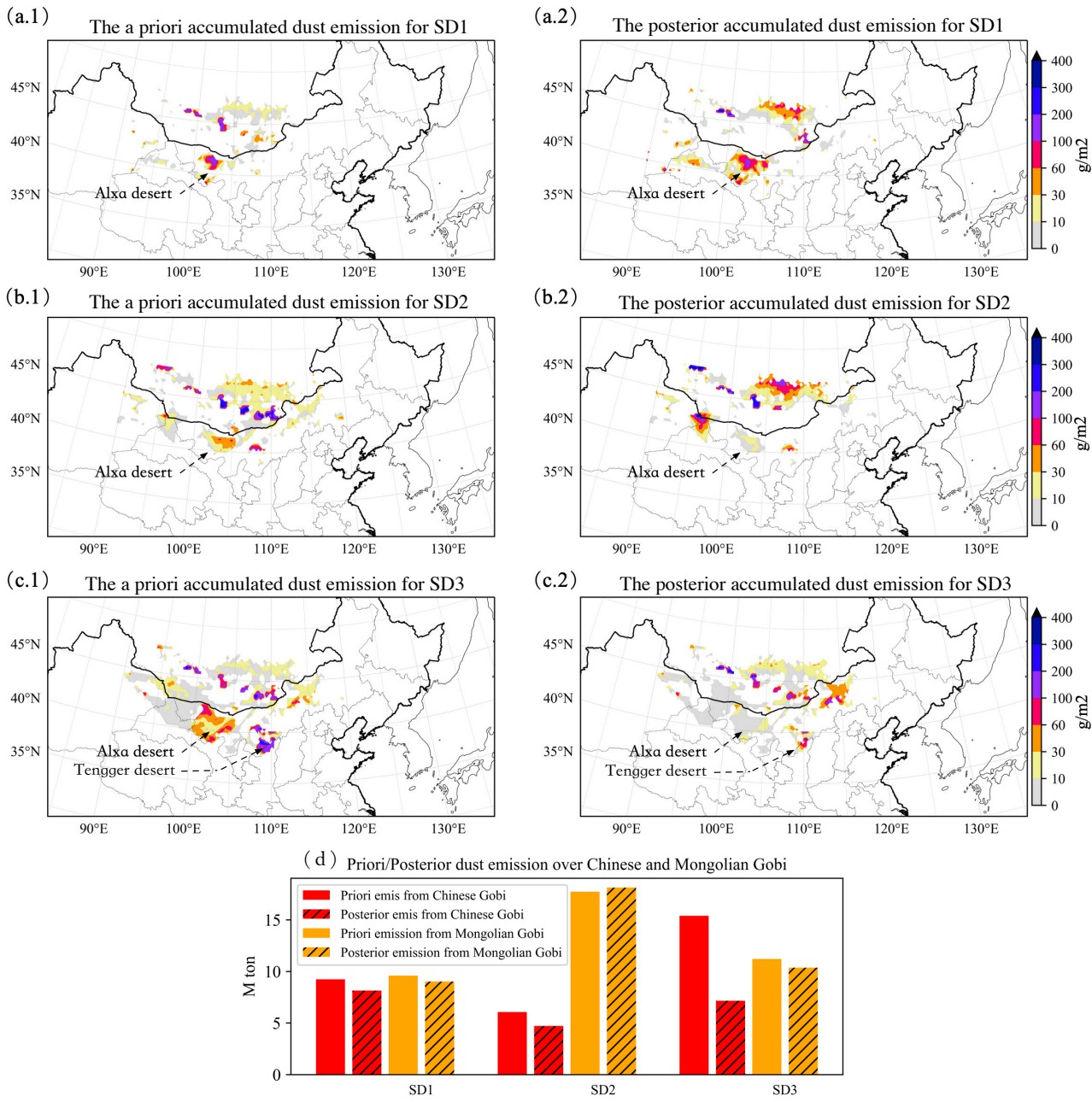

**Figure 6.** Distribution of the a priori (a.1-c.1) and posterior (a.2-c.2) accumulated dust emission for the SD1, SD2 and SD3; the total priori and posterior emission either from China or from Mongolia during SD1, SD2 and SD3 (d).

A snapshot of the a priori and posterior surface dust concentration and AOD during SD1 are shown in the middle and right panel in Figure 2. Both of the a priori and posterior simulations show a similar pattern as the (bias-corrected) $PM_{10}$ and AOD observations that are shown in Supplementary Figure S1 and Figure S2, or the raw data shown in the left panel of Figure 2. However, the dust concentrations are underestimated in the a priori simulation in the entire region, resulting in a $PM_{10}$ RMSE

of 833 $\mu g/m^3$, and AOD RMSE of 1.36 and 1.53. Compared to the a priori simulation, the posterior emission field simulated a more severe dust plume, and the $PM_{10}$ RMSE is reduced to 743 $\mu g/m^3$, and the AOD RMSE declined to 1.30 and 1.34, simultaneously. One important reason for this high error residue is the position mismatch among the simulation and observations. For instance, the plume front (red dash line) that is visible in the MODIS AOD retrievals in Figure 2(b.1) is about 100 km ahead of the front line shown in $PM_{10}$ measurements in Figure 2(a.1). It is possible that the dust plume in the higher layers moved

faster, and was in further southeast than the dust cloud at bottom layer. However, this feature is not correctly captured by our LOTOS-EUROS/dust model. Both the simulated plume of AOD and surface dust concentration are in the same position; the simulated plume fronts indicated in Figure 2(a.2) and (b.2) moved faster than the front line indicated by $PM_{10}$ measurements, but slower than the front line in the MODIS AOD. The mismatch in simulated vertical structure is mostly likely caused by uncertainty in the advection transport. The two-dimensional grid distortion technique (Jin et al., 2021) which is independent

on the emission inversion could adjust the horizontal position of the dust cloud simulation to better fit the available measurements, but is not yet able to adjust the vertical structure as is required here. A three-dimensional grid distortion with data that measuring the vertical profile of dust cloud is planned to solve this issue in our future research.

Figure 3 shows snapshots of original AOD and $PM_{10}$ measurements, as well as the a priori and posterior AOD and surface dust concentration simulation at March 28, 11:00 (CST) within SD2. What stands out in the *a priori* AOD simulation in Figure

3(a.2) is the overestimation, especially, in the center of the plume in the NCP region. The standard model simulated AOD values even larger than 4, while the measurements were around 2 to 3. It should be noted that all the AOD measurements shown in Figure 3(a.1) contain both the dust and non-dust fraction, and the baseline-removed AODs which can be found in Figure S4 are therefore a bit lower. By assimilating these bias-corrected measurements, posterior AOD simulation in Figure 3(a.3) are now in better agreement with the AOD observations. The AOD RMSE is therefore reduced from 0.99 in the *a priori*

simulation to 0.81 in the *posterior* simulation. The improvement is also seen in the surface $PM_{10}$ simulation, where the RMSE score decreases from 471 $\mu g/m^3$ in the *a priori* to 359 $\mu g/m^3$ in the *posterior* simulation.

Scenes of original AOD and $PM_{10}$ measurements, the a priori and posterior simulated AOD and surface dust concentration at April 15, 11:00 (CST) during SD3 are plotted in Figure 4. It is apparent from this figure that dust concentrations are overestimated by model within this event as well. For a case in point, the $PM_{10}$ measurements in Figure 4(a.1) show that

FWP is almost free of dust at that moment, and this is confirmed by the baseline-removed $PM_{10}$ measurements shown in Supplementary Figure S5. The multi-observation assimilation successfully resolves the $PM_{10}$ and AOD measurements. The simulation driven by the posterior emission field is in much better agreement with the measurements, with the $PM_{10}$ RMSE is reduced from 891 $\mu g/m^3$ to 144 $\mu g/m^3$, and the AOD RMSE drastically decreases from 1.79 to 0.73.

## 4.2 AOD-only or PM$_{10}$-only assimilation evaluation

Next to the multi-observation (AOD and PM$_{10}$ together) emission inversion described in Section 4.1, assimilation tests with same configurations but using only AOD or PM$_{10}$ observations are carried out as well. They are referred to as *AOD-only* and *PM$_{10}$-only* assimilation in this study. Difference and added value of the multi-observation assimilation vs. AOD-only/PM$_{10}$-only assimilation are analyzed. In addition, once AOD or PM$_{10}$ measurements are assimilated solely, the other one will be used as the independent data for validation.

Figure 7 shows snapshots of the dust AOD and surface concentration simulation driven by the AOD-only (panel 2) and PM$_{10}$-only (panel 3) posterior emission field with the SD3. It clearly illustrates the typical results that are observed in the other two events. As aforementioned, the AOD RMSE is reduced from 1.79 (the a priori) to 0.72 in the multi-observation assimilation. Once the MODIS AODs are assimilated solely, better dust AOD simulation performance is obtained with the AOD RMSE further reduced to 0.69. The effectiveness of emission estimation through assimilating AOD is validated using the independent PM$_{10}$ measurements, the PM$_{10}$ RMSE decreased from 891 $\mu$g/m$^3$ (the a priori) to 210 $\mu$g/m$^3$ simultaneously. However, the PM$_{10}$ RMSE stays at a bit higher level compared to 143 $\mu$g/m$^3$ which is obtained in the multi-observation assimilation. Similarly, the PM$_{10}$-only assimilation provides a further lower PM$_{10}$ RMSE 133 $\mu$g/m$^3$, but the simulated AOD field (RMSE = 0.77) is not as accurate as the one from the multi-observation assimilation.

Scenes of dust AOD and surface concentration simulation from the AOD-only and PM$_{10}$-only assimilation in SD1 and SD2 can be found in Figure S7 and Figure S8 in the Supplement. The corresponding AOD and PM$_{10}$ RMSEs are calculated and shown in Table 2. Very similar trends are found in the assimilation tests within SD2. For SD3, the AOD-only or PM$_{10}$-only assimilation also results in closer dust simulation to the assimilated measurements. However, poor performance (compared to the a priori) is obtained against the independent observations. For instance, driven by the emission through assimilating the AOD, the posterior simulated dust concentration are very different from the PM$_{10}$ measurements, and the RMSE is lifted up to 887 $\mu$g/m$^3$ from 833 $\mu$g/m$^3$ (the a priori). It is mainly because that the AOD and PM$_{10}$ measurements indicate the different dust plume position in the simulation space as described in Section 4.1.2, and therefore the assimilation would lead the dust simulation into mismatch with the independent one.

Therefore, any AOD-only or PM$_{10}$-only assimilation would only result in the posterior closer to the assimilated data, but the posterior simulation in the independent observation space is not ensured to be improved. It is even possible that the posterior is misled when the dust vertical profile is not well reproduced (such as our simulation over SD1) or systematical observation bias is present. In this case, multi-observation assimilation used in this study is a safe choice to avoid the model divergent.

## 4.3 Source apportionment

China government has launched several large-scale ecological engineering projects to combat the environmental problems in the northern China during recent decades. One of the largest is the Three-North Shelter Forest Program, which aimed at increasing the vegetation cover upto 15% by 2050 (Niu et al., 2019). Several studies (Shao et al., 2013; Tan and Li, 2015) reported that the vegetation recover weakened dust storms substantially. In contrast, Mongolia has experienced the ever-increasing land

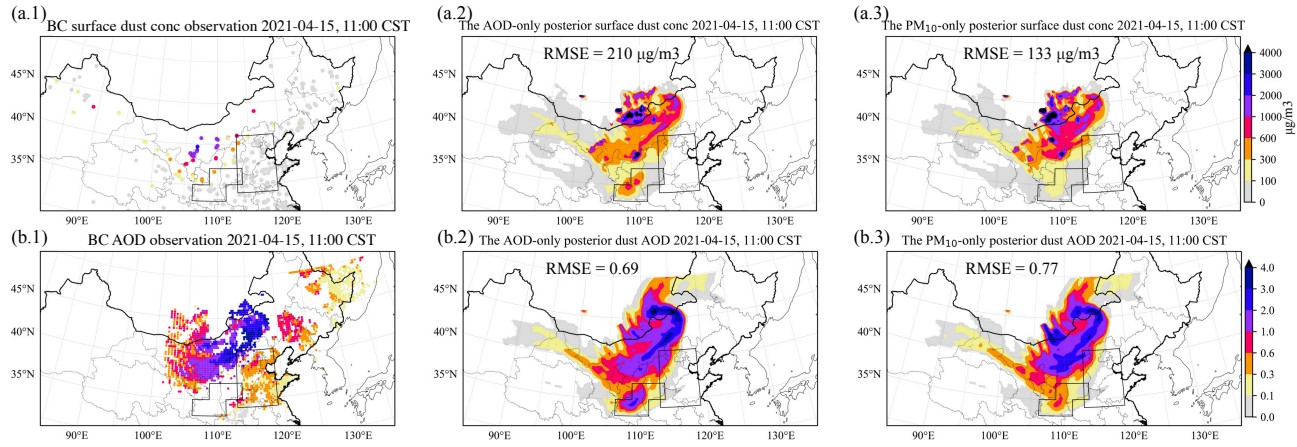

**Figure 7.** The posterior dust surface dust concentration and AOD simulation either driven by the AOD-only or driven by the $PM_{10}$-only assimilation emission result during SD3. Column 1: the bias-corrected (BC) $PM_{10}$ and AOD observations; column 2: the dust surface concentration and AOD simulation driven by the posterior emission from AOD-only assimilation; column 3: the dust surface concentration and AOD result driven by the posterior emission from $PM_{10}$-only assimilation at 11:00, April 15.

**Table 2.** Evaluation (RMSE) of the posterior dust simulation either by assimilating AOD, or by assimilating $PM_{10}$, or by assimilating both of them.

|  | Timeline | Priori | Multi-observation | AOD only | $PM_{10}$ |
|---|---|---|---|---|---|
| SD1 | $PM_{10}$ at 10:00, Mar 15 | 833 $\mu g/m^3$ | 743 $\mu g/m^3$ | 887 $\mu g/m^3$ | 692 $\mu g/m^3$ |
|  | AOD at 10:00, Mar 15 | 1.36 | 1.30 | 1.31 | 1.66 |
|  | AOD at 12:00, Mar 15 | 1.53 | 1.34 | 1.26 | 1.63 |
| SD2 | $PM_{10}$ at 11:00, Mar 28 | 471 $\mu g/m^3$ | 359 $\mu g/m^3$ | 366 $\mu g/m^3$ | 351 $\mu g/m^3$ |
|  | AOD at 11:00, Mar 28 | 0.99 | 0.81 | 0.79 | 0.89 |
| SD3 | $PM_{10}$ at 11:00, Apr 15 | 891 $\mu g/m^3$ | 143 $\mu g/m^3$ | 210 $\mu g/m^3$ | 133 $\mu g/m^3$ |
|  | AOD at 11:00, Apr 15 | 1.79 | 0.72 | 0.69 | 0.77 |

degradation and desertification (Meng et al., 2020), which aggravates the spring dust storms (Han et al., 2021). To evaluate the roles of Mongolian and Chinese Gobi deserts in the 2021 super sandstorms quantitatively, source apportionment tests based on the estimated emission field are carried out further. These source apportionment tests focus on the two dust-affected mega-city clusters in the northern China, North China Plain (NCP) and Fenwei Plain (FWP), and aims to calculated whether the dust originates from transnational transport from Mongolia or from domestic sources in China.

Two LOTOS-EUROS/dust simulations have been conducted with the posterior emission field obtained in the multi-observation assimilation in Section 4.1.1, but with either only the emissions in Mongolia or the emissions in China enabled. As can be seen in the time series of the hourly $PM_{10}$ concentration measurements in NCP and FWP Figure 1(c-d), after the peak of the dust past by the NCP and FWP region, they still suffered from some less severe dust affection. Therefore, longer simulation windows are

used in these source apportionment tests to simulate the full life cycle of these dust events as can be seen in Table 1. Different from cases caused by other pollutants, the spring dust events in East Asia are usually short-term events, with concentrations of dust that quickly increase to huge levels, but also drop down quickly after the storm passed by. Metrics as daily average dust concentration do not reflect the dust intensity directly. Therefore, a dust deposition index is introduced that measures the sum
of dry and wet deposition to quantify the impact of the dust in the studied regions.

Figure 8 shows the spatial pattern of the deposition for dust originating from China (left panels) and originating from Mongolia (right panels) in SD1, SD2 and SD3. The most interesting finding is that the sources in Mongolia play a much more important role in dust pollution in northern China than the sources in China itself. As shown in Figure 8 (a.1), a huge amount of particles were released in the Alxa desert during SD1, but these were mainly transported westward and only a little fraction
of them moved to the densely populated areas. During SD2, the deposition of dust released from China is non-eligible, but the total deposition is still dominant by dust released from Mongolia. Within SD3, the dust particles emitted from Chinese Gobi were spread all over the northern China and hence played a more significant role.

The total deposition in the NCP and FWP regions has been calculated and is shown in Figure 8(d). For the NCP region, 81k, 118k, and 70k tons of Mongolian dust were deposited during SD1, SD2, and SD3 events, while the total deposition
from Chinese desert was about 8.3k, 20k, and 93k tons, respectively. For the other important cluster FWP, 4.3k, 22k, and 24k tons were attributed to domestic sources, and 20k, 57k, and 7.5k tons of dust were attributed to transnational transport from Mongolia. In general, Mongolian Gobi pose more severe threat to FWP and NCP region than the Chinese Gobi. About 63% of the dust deposition in FWP is attributed to the transnational transport. Over NCP, this value further grows up to 69%.

## 5   Summary and future work

In spring 2021, three super dust storms occurred in East Asia after being absent for one and a half decades, which brought enormous health damages and property losses. To exploit the reappeared super sandstorms, inverse modeling was conducted through optimizing the dust simulation with observations of MODIS AODs and ground-based $PM_{10}$ concentration from the Chinese MEP air quality monitoring network. Data quality controls were designed and applied in order to use the AOD and $PM_{10}$ measurements for representing the dust load. Based on the most likely emission field calculated by the inversion, source
apportionment was further performed to derive the contribution of transnational transport from Mongolia and domestic dust emission to the dust pollutant level in northern China.

Emission inversion was successfully performed by assimilating the AOD and $PM_{10}$ concentration measurements. The multi-observation assimilation showed that windblown dust emission occurred actively both in Chinese and Mongolian Gobi deserts during the events studied. Overall, about 37.5M tons of dust was released in Mongolia, and the total emission from the Chinese
Gobi was also as high as 19.9M ton. The simulated AOD and surface dust concentration driven by the posterior emission fields have been validated to be in better agreement with the observations. To obtain a further accurate dust field analysis, however, vertical structure adjusting techniques such as 3D grid distortion are then in demand for fully resolving the ground-based $PM_{10}$ and column-integrated AOD together, and will be explored in our future work.

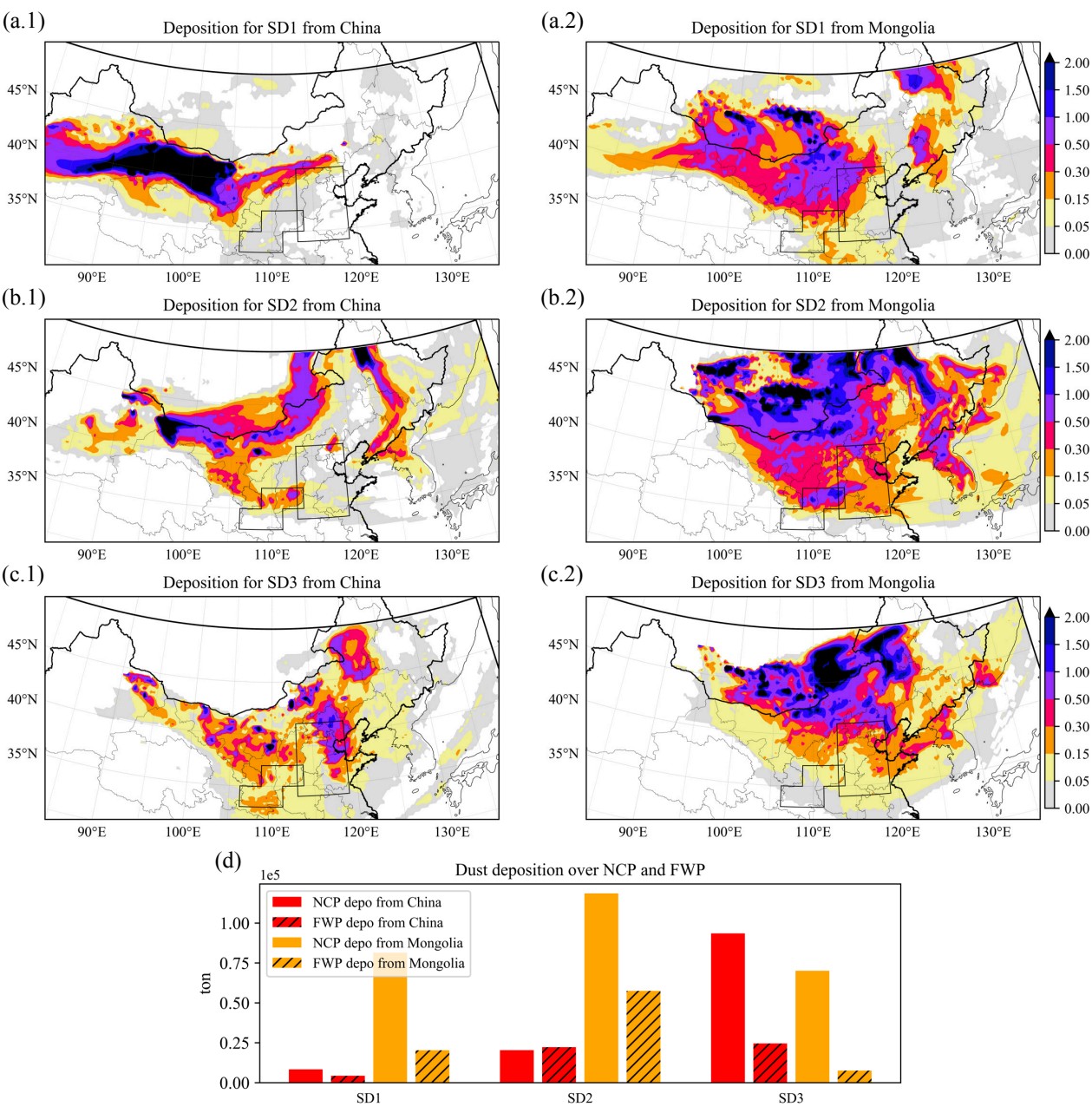

**Figure 8.** Spatial patterns of deposition of dust originating from China (a.1-c.1) and from Mongolia (a.2-c.2). Panel (d) shows the total mass of dust deposition on NCP and FWP either from China or from Mongolia during SD1, SD2 and SD3 (d).

Emission inversions that only assimilated the AOD or $PM_{10}$ concentration were also carried out. The comparison against the multi-observation assimilation showed that AOD-only or PM10-only assimilation would result in the posterior further closer to the assimilated data, but the improvement of the simulation in the independent observation space is not ensured. Especially when vertical structure is not well reproduced (such as the simulation over SD1), the posterior would be divergent. Under this

circumstance, multi-observation assimilation used in this study is the more reliable choice.

A source apportionment study was then performed based on the multi-observation assimilation estimated emission by estimating the origin of the dust that was deposited in regions in northern China. It indicated that Mongolian Gobi posed more severe threats to Fenwei Plain (FWP) and North China Plain (NCP) than Chinese Gobi within the three 2021 spring dust storms. For FWP, about 63% of the dust deposition originated from transnational transport from Mongolia. In NCP, the Mon-

golian dust contribution was also as high as 69%. To further explore the roles of specific deserts (such as Alxa and Tengger deserts in Chinese Gobi) and long-distance transport patterns on the dust affection in the northern China, more complex source apportionment tests are planned in our future research.

### Code and data availability

The source code and user guide of the LOTOS-EUROS CTM could be obtained from https://lotos-euros.tno.nl. The real-time

$PM_{10}$ data are from the network established by the China Ministry of Environmental Protection and available to the public via http://106.37.208.233:20035/ (Internet Explorer 10 or 11), the $PM_{10}$ data used in this paper is also archived on Zendo (https://doi.org/10.5281/zenodo.6459866). The MODIS Deep Blue C6 data suites are available at https://ladsweb.modaps.eosdis.nasa.gov/. The datasets including measurements and model simulations can also be accessed by contacting the corresponding author.

### Acknowledgments

This work is supported by the National Natural Science Foundation of China [grant 42105109] and Natural Science Foundation of Jiangsu Province (NO.BK20210664).

We would like to thank the referee Sekiyama Thomas for the in-depth comments.

### Author contribution

JJ conceived the study and designed the dust storm data assimilation. JJ, MP and AS performed the control and assimilation tests and carried out the data analysis. AS, HL, WH, BL, and HXL provided useful comments on the paper. JJ prepared the manuscript with contributions from AS and all others co-authors.

**Competing interests**

The authors declare that they have no conflict of interest.

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
