# Peer review of "Inverse modeling of the 2021 spring super dust storms in East Asia"

_Atmospheric Chemistry and Physics, 2021_

## Author Comment (AC1)

**Response to Referee #1:** We would like to thank the referee for the careful review throughout the paper and the useful comments. Especially the suggestion for comparison of multi-observation against $PM_{10}$-only/AOD-only assimilation and replacement of "local source" with "domestic source" , it helped us to improve the quality of the manuscript.

Our response follows (*the reviewer's comments are in italics and blue*)

*General comments:*

*This work explores the dust emission and deposition distributions in East Asia induced by an extremely severe dust storm in spring 2021 using the four-dimensional variational method. The authors used scientifically plausible experimental methods with reasonable observation datasets. Therefore, this manuscript will be acceptable as a paper with minimum quality standards after a major revision of English errata and a minor revision of scientific descriptions. However, the experimental designs and conclusion of this work are not scientifically significant very much. To increase the scientific significance of this work, the authors should carry out additional data-assimilation experiments and reclassify dust-source regions in China/Mongolia. The reclassification of dust-source regions could change the authors' conclusion. The decision to accept or reject this manuscript as an article of Atmospheric Chemistry and Physics, one of the high-impact journals, is left to the editor.*

*Major comments:*

*Page 12, Lines 17-18: generally, the official uncertainty of remote-sensing and in-situ measurements includes only instrumental errors. It doesn't include data-screening errors and spatial representative errors, while 0.25x0.25-degree grid dust models needs large representative errors. The official AOD uncertainty of the MODIS Deep Blue product is probably underestimated especially under cloudy conditions. The assumption of the PM10 observation error (10% in Jin et al. 2018) is also underestimated due to the lack of spatial representative errors. Plus, bias-correction errors are not included in this study. That means that the weight of observations will be overestimated in the data assimilation system. This might be one of the reasons that the priori and posteriori emission distributions are very different in this study.*

**Reply**: Thanks for point out the important source of the observation uncertainty/error. We only calculated the instrument error and overlooked the representation one. To make the $PM_{10}$

concentration and MODIS AOD observations fully representative, we now re-design the covariance of AOD and $PM_{10}$ measurements.

Concerning the uncertainty of $PM_{10}$, we used the max(200, 10%*$y^{AOD}$) to characterize the uncertainty of the $PM_{10}$, instead of using 10% choice simply. Of course, this was not fully explained. Besides, we agree with the referee that uncertainty in the non-dust bias correction should be taken into account. Details concerning the observation error covariance are added in page 13 line 23-31 and page 14, line 1-7 by saying "***Both the instrument and representing errors are considered when the observation error covariance, $O^{PM}$ and $O^{AOD}$, are designed. The uncertainty (square root of the individual diagonal element in $O^{PM}$) of the pre-processed $PM_{10}$ measurements for assimilation is assumed to be due to uncertainty in the $PM_{10}$ data and the non-dust $PM_{10}$ bias correction. We have used $\sigma^{PM}$ = max( 200, 10%·$y^{PM}$+180 ) to characterize the uncertainty of $PM_{10}$ data. It follows the choice of 10% in our previous study (Jin et al., 2018) with uncertainty inflated for this application. This is mainly to prevent the posterior from getting too close to the low-value $PM_{10}$ observations and hence being model divergent. In addition, the uncertainty of the non-dust $PM_{10}$ simulation $\sigma^{BC}$ that is introduced in Section 2.2 is set to 40% following the aerosol simulation analysis over China using LOTOS-EUROS (Timmermans et al., 2017). The integrated uncertainty $\sigma^{integrated}$ for using the bias-corrected $PM_{10}$ to represent the dust load is then calculated as:***"

$$\sigma^{\text{integrated}} = \{ \left(\sigma^{\text{PM}}\right)^2 + \left(\sigma^{\text{BC}}\right)^2 \}^{0.5} \tag{11}$$

***Snapshots of $\sigma^{PM}$ and $\sigma^{integrated}$ distribution accompanying the $PM_{10}$ measurements shown in Figure 2(a.1), Figure 3(a.1) and Figure 4(a.1) in the three dust events are shown in Figure S1(c-d), Figure S3(c-d) and Figure S5(c-d), respectively.***

[Figure]

*Fig.S1 Snapshots of the bias-corrected PM₁₀ measurements for assimilation during severe dust event 1 (SD1) at March 15, 12:00. a.1: the non-dust PM₁₀ simulation simulated by LOTOS-EUROS, b.1: the bias-corrected PM₁₀ measurements, c.1 and d.1: the instrument uncertainty and integrated uncertainty of the PM₁₀ observations.*

*The integrated uncertainty of AOD measurements for assimilation is also calculated as the sum of the instrument error and the error of the non-dust AOD bias correction. The former is taken directly from the MODIS Deep Blue product, while the uncertainty of non-dust AOD simulation is set to 40% as well. Snapshots of the AOD instrument uncertainty and integrated uncertainty with respect to the AOD observations shown in Fig.S2, Fig.S4 and Fig.S6.*

[Figure]

*Fig.S2 Snapshots of the bias-corrected dust AOD measurements for assimilation during severe dust event 1 (SD1) at March 15. a.1: the non-dust AOD simulation simulated by LOTOS-EUROS, b.1: the bias-corrected AOD measurements, c.1 and d.1: the instrument uncertainty and integrated uncertainty of the AOD observations at 10:00 CST. Similar, a.2-d.2 show the non-dust AOD simulation, the bias-corrected AOD and the AOD uncertainties at 12:00.*

The emission inversions are re-conducted using the new observation error covariance, and the source apportionment tests are repeated with the newly updated emission fields.

*Page 15, Lines 20-21: the authors say "a three-dimensional grid distortion should be developed to solve this issue," but I don't agree with it. Only dust emissions are control variables in this study. Advection cannot be controlled. Therefore, the inconsistency in the wind field should be tried to solve first. Otherwise, the dust data assimilation system has to keep using the wind field data and advection model that cannot reproduce vertical sheers to inverse dust emissions. This inconsistency is first and foremost a matter of the model dynamics.*

**Reply:** The grid distortion technique was of course not well described. In fact, the grid distortion data assimilation (Jin et al., 2021) is independent on the dust emission inversion used in this paper. It is capable of adjusting the position of the simulated dust plume which is likely to be caused by the advection transport as the reviewer indicated. The re-position is carried out by distorting the model coordinates instead of adjusting the advection. The grid distortion was used in our previous work (Jin et al., 2021) to re-align the simulated 2D dust plume to best fit the ground $PM_{10}$ data phase. The relation between the emission inversion and grid distortion can been best seen from the diagram figure below.

[Figure]

Figure 6. Diagrams of emis inversion, grid-distorted assim and hybrid assim systems, from Jin et al., 2021

For the 3D vertical structure mismatch in this paper, however, 3D grid distortion data assimilation with data that measuring the vertical profile are required. To make it clear, remarks are added in page 17 line 11-17 by saying "***Both the simulated plume of AOD and surface dust concentration are in the same position; the simulated plume fronts indicated in Figure 2(a.2) and (b.2) moved faster than the front line indicated by PM₁₀ measurements, but slower than the front line in the MODIS AOD. The mismatch in simulated vertical structure is mostly likely caused by uncertainty in the advection transport. The two-dimensional grid distortion technique (Jin et al., 2021) which is independent on the emission inversion could adjust the horizontal position of the dust cloud simulation to better fit the available measurements, but is not yet able to adjust the vertical structure as is required here. A three-dimensional grid distortion with data that measuring the vertical profile of dust cloud is planned to solve this issue in our future research.***"

*Page 16, Lines 10, 29 and 31; Page 18, Lines 6 and 23: it's misleading to refer to domestic sources/emission/deserts as "local" ones. Local means neighborhood or specific. In this study, for instance, the distance from Beijing City in NCP to the Alxa Desert in China is more than 1000 km while it's 550 km to the Sino-Mongolian border. However, the authors refer to Alxa as a local source but Mongolian Gobi as a faraway source. The "local" should be replaced by "domestic".*

**Reply**: "***domestic***" is indeed more suitable than "***local***", we have corrected it throughout the paper.

*Page 16, Lines 21-25: the authors found interesting characteristics of dust transport and deposition for each storm period. This study would have been a much better paper if the authors gave a deeper insight into the meteorological causes of the dust transport/deposition characteristics rather than just stating the facts. Each transport/deposition can be probably explained by the dynamics of synoptic meteorological fields.*

**Reply**: This paper focuses now on the emission inversion through assimilating multiple observations. The estimated emission field is useful. One application as shown in this paper is that it helps to evaluate the emission intensity in different regions and their contribution to dust affection in the densely populated areas. We agree that how the synoptic meteorology governs the way of long-distance transport are interesting, it would be explored in our future research.

Remarks are added in page 22, line 10-12 by saying "***To further explore the roles of specific deserts (such as Alxa and Tengger deserts in Chinese Gobi) and long-distance transport patterns on the dust affection in the northern China, more complex source apportionment tests are planned in our future research.***"

*In addition, I have a question. Did the source apportionment simulations last for only the SD1, SD2, or SD3 period? If so, the simulation periods are only 2 or 3 days. When a dust plume flows directly from the source region to NCP or NWP, the 2 or 3 days is long enough. However, when a dust plume is caught by synoptic disturbances multiple times, it might take more than 3 days for the plume to travel 2000 km from western Mongolia to NCP/NWP. Is it possible that the short simulation time is one of the reasons why the Alxa Desert less influenced NCP/NWP during SD1?*

**Reply:** Thanks for pointing out this issue. The same simulation window was used for the source apportionment tests. However, the simulation window seems not long enough, there is still some less severe dust affection out of the window. Therefore, we now repeat all the source apportionment tests with new simulation periods. Remarks are added to explain how we choose the new simulation window in page 19, line 6-10, ***"Two LOTOS-EUROS/dust simulations have been conducted with the posterior emission field obtained in the multi-observation assimilation in Section 4.1.1, but with either only the emissions in Mongolia or the emissions in China enabled. As can be seen in the time series of the hourly $PM_{10}$ concentration measurements in NCP and FWP Figure 1(c-d), after the peak of the dust past by the NCP and FWP region, they still suffered from some less severe dust affection. Therefore, longer simulation windows are used in these source apportionment tests to simulate the full life cycle of these dust events as can be seen in Table 1."***

Discussion concerning contribution of Chinese Gobi and Mongolia Gobi to the dust deposition over NCP and FWP region is updated in Section ***4.2 Source apportionment.***

[Figure]

**Figure 1. (a) Distribution of the potential dust emission source (barren and sparse vegetation landcover) over East Asia and the China MEP observing network over northern China; (b) The spring (March-May) mean PM10 concentration observations over NCP and FWP from 2019 to 2021; Time series of the hourly PM10 reported by stations in NCP (c) and FWP region (d) during SD1 (column 1), SD2 (column 2) and SD3 (column 3).**

**Table 1.** Descriptions of the three severe dust storm events that occurred in China in Spring 2021. Timezone is China Standard Time (CST))

| dust event | affected regions | highest PM$_{10}$ [$\mu g/m^3$] | assimilation window | source apportionment simulation timeline |
|---|---|---|---|---|
| SD1 | NCP | 9993 | March 13, 00:00 to March 15, 23:00 | March 13 to 17 |
| SD2 | NCP, FWP | 9985 | March 26, 00:00 to March 28, 23:00 | March 26 to 29 |
| SD3 | NCP, FWP | 4113 | April 14, 00:00 to April 15, 23:00 | April 14 to 16 |

*Page 18, Summary and conclusion:*

*Point 1: The authors repeatedly emphasized that the MODIS AODs were screened by Angstrom exponents and bias-corrected by non-dust aerosol simulations. The PM10 data were also bias-corrected. The authors cited the papers of these preprocessing methods, but didn't present the improvements of the inversion for the 2021 dust storms made by these preprocessing methods at all. Even if the preprocessing methods worked well in the case of previous studies, there*

*might be not much positive impact on the inversion of the 2021 dust storms. If the preprocessing methods are emphasized in Abstract and Summary, it should be shown in this manuscript how much the inversion is improved by the preprocesses for the cases of the 2021 dust storms. If not, the preprocessing methods shouldn't be emphasized in Abstract and Summary.*

**Reply**: Accepted. ***Abstract*** and ***Summary*** are updated. Data quality controls are not emphasized here.

*Furthermore, the authors emphasized that both the MODIS AOD and PM10 data were simultaneously assimilated to estimate dust emissions. However, its benefits were not quantitatively presented in this study. I'm very interested in the difference of the inversion results between a MODIS AOD-only assimilation, a PM10-only assimilation, and the simultaneous assimilation. If the difference is shown in the manuscript, it will be an alternative to independent validation with a subset of leaving data. This study would have been a much better paper if the authors presented more than one inversion results illustrating the quantitative improvements made by the preprocessing methods and the simultaneous data assimilation.*

**Reply:** Thanks for the suggestion. The $PM_{10}$-only and AOD-only assimilations are indeed interesting and now performed. In most cases (SD2 and SD3) of the $PM_{10}$-only and AOD-only assimilation tests, the improvement on the dust simulation can be validated through the comparison against the independent measurements that are not assimilated.

In addition, we also compare the multi-observation assimilation against the PM10-only and AOD-only ones. It indicates any AOD-only or $PM_{10}$-only assimilation would only result in the posterior closer to the assimilated data, but the posterior simulation in the independent observation space is not ensured to be improved. It is even possible that the posterior is misled when the dust vertical profile is not well reproduced (such as our simulation over SD1). In this case, multi-observation assimilation used in this study is a safe choice to avoid the model divergent. A new subsection "***4.2 AOD-only or PM10-only assimilation evaluation***" in page 18 is now added to describe the AOD-only and $PM_{10}$-only assimilation result.

*Point 2: The authors divided the dust source regions into Chinese sources and Mongolian sources in this study. However, it's not a scientific classification because the Sino-Mongolian border in the Gobi Desert was artificially or politically drawn, not geologically or biologically. Although the authors cited Han et al. (2021) for Mongolian desertification, Han et al. (2021)*

*evaluated only Mongolia and didn't compare the Mongolian Gobi and the Chinese Gobi quantitatively. If the classification with the Sino-Mongolian border is a scientific or environmentally crucial issue, it should be clarified first that the desertification in the Mongolian Gobi is much more serious than in the Chinese Gobi, before the source apportionment study.*

**Reply: Definitely, we should explain that why we analyze the contribution of Mongolian dust and Chinese dust. Remarks are added in page 18, line 30-33 and page 19, line 1-5, by saying** ***"China government has launched several large-scale ecological engineering projects to combat the environmental problems in the northern China during recent decades. One of the largest is the Three-North Shelter Forest Program, which aimed at increasing the vegetation cover upto 15% by 2050 (Niu et al., 2019). Several studies (Shao et al., 2013; Tan and Li, 2015) reported that the vegetation recover weakened dust storms substantially. In contrast, Mongolia has experienced the ever-increasing land degradation and desertification (Meng et al., 2020), which aggravates the spring dust storms (Han et al., 2021). To evaluate the roles of Mongolian and Chinese Gobi deserts in the 2021 super sandstorms quantitatively, source apportionment tests based on the estimated emission field are carried out further. These source apportionment tests focus on the two dust-affected mega-city clusters in the northern China, North China Plain (NCP) and Fenwei Plain (FWP), and aims to calculated whether the dust originates from transnational transport from Mongolia or from domestic sources in China."***

*Furthermore, the authors concluded that "local" Chinese deserts play a small role in the dust deposition over FWP and NCP compared to the contribution of "long" transports from Mongolia. This conclusion is very misleading. The northeastern part of the Gobi Desert, which is located in Mongolia, is much closer to FWP/NCP than the southwestern part of the Gobi Desert, which is located in China. In other words, Chinese deserts are not always "local" for FWP/NCP and Mongolian deserts are not always far away from FWP/NCP. I think it will be more scientifically plausible to classify dust source regions using the distance, latitude, altitude, and vegetation, not using national borders. If the reclassification was performed, this study would be an excellent paper.*

**Reply**: We agree with the referee that "local source" is a bit misleading because the Chinese Gobi is also far away from the NCP region. Descriptions about the source apportionment in Summary in page 22, line 6-12 is now updated as:

*"A source apportionment study was then performed based on the multi-observation assimilation estimated emission by estimating the origin of the dust that was deposited in regions in northern China. It indicated that Mongolian Gobi posed more severe threats to Fenwei Plain (FWP) and North China Plain (NCP) than Chinese Gobi within the three 2021 spring dust storms. For FWP, about 63% of the dust deposition originated from transnational transport from Mongolia. In NCP, the Mongolian dust contribution was also as high as 69%. To further explore the roles of specific deserts (such as Alxa and Tengger deserts in Chinese Gobi) and long-distance transport patterns on the dust affection in the northern China, more complex source apportionment tests are planned in our future research."*

*Specific comments:*

*Figures 2-5 and S1-S3: The font of each map's title is too small.*

**Reply:** Modified.

*Page 6, Lines 15-16: I don't think the bias-corrected PM10 data CLEARLY shows the shape of the dust storm. It's too subjective to say "clear" for this distribution.*

**Reply**: "clearly" is removed.

*Page 6, Line 17: the authors say "the shape of the simulated dust plume matches well with the observed shape," but I think "it's slightly matched." To use "well" is overvaluation.*

**Reply**: "Well" is removed.

*Page 8, Line 25: the model has only eight layers from the surface to 10 km, which is very sparse, especially in the PBL, to investigate aerosol emission and deposition. Usually, state-of-the-art aerosol dispersion models have more than 50 layers from the surface to the tropopause, including more than 10 layers only in the PBL. Why is the vertical resolution set so low? Even if the meteorological fields are provided from ECMWF, the sparse layers in the dust model will result in a very large vertical numerical diffusion, which deteriorates regional aerosol simulations. May I have the authors' opinion?*

**Reply:** The currently used vertical resolution is indeed rather course. The main reason for using the chosen configuration is that it is the same as what was used for the full chemistry and aerosol simulations from which the non-dust contribution to particulate matter in observation sites was computed. Since these are expensive calculations, especially over longer time periods, the

number of layers in these simulations is limited. For future studies an increase in vertical resolution is considered for at least the dust simulations. Especially in case observations could be used that provide more information on the vertical structure than currently used surface PM and total column AOD, for example satellite aerosol layer height observations, it will be useful use the highest possible vertical resolution.

*Page 12, Eq. 4: there's no explanation for the background error covariance matrix B in the text.*

**Reply**: Explanation for the background error covariance **B** is definitely important. We now rewrite Section "***3.1 Assimilation method***" with detailed descriptions about the background error covariance **B** and observation error covariance $\mathbf{O_{PM}}$ and $\mathbf{O_{AOD}}$. Details could be found below or in page 12-14.

[revised manuscript text omitted]

*Page 12, Line 9: This study built a background error covariance matrix using ensemble simulations. If the tangent linear method used in this study resembles ensemble-variational (EnVar) methods, how to prepare the ensemble perturbations is critical for the assimilation*

*performance. Although Jin et al. (2018) briefly describes how to make the perturbations, this important issue should be described in detail here.*

**Reply**: See reply to the previous question.

*Page 13 Line 9: I don't think the name of Alxa desert is famous outside of China. Usually, the area is considered as a part of the Gobi Desert. Please indicate its location in Fig. 1 with a large font, not only in Fig. 6.*

**Reply:** Figure 1(a) is updated as can be seen above or in page 6. Remarks is now added in page 14, line 26-27 by saying "***The a priori model simulation indicates dust emission took place over both the Alxa desert (part of Chinese Gobi).......***"

*Page 13, Line 10, Line 15, Line 21: maximum emissions per unit area are not very meaningful because they strongly depend on the horizontal model resolution.*

**Reply:** The exact value of the maximum emission is indeed not very meaningful, it may change a lot when different spatial resolution is used. "***with a maximum of 221 g/m2 in China and 286 g/m2 in Mongolia***" is now modified to "***their maximum emission flux exceeded 200 g/m2***" in page 14, line 27-28; "***the maximum emission accumulation here is as high as 370 g/m2***" is now changed to "***The emission accumulations in several grid cells here are in the order of 300 g/m2***" in page 15, line 4; "***the accumulated emissions reach values up to 270 g/m2***" is also changed to "***the accumulated emissions reach values over 200 g/m2***" in page 15, line 11.

*Page 15, Lines 10-11: "resulting in a RMSE of 833 ug/m3, 1.36 and 1.53" this phrase is very hard to understand at first glance because it's not easy to realize 1.36 and 1.53 are AODs.*

**Reply:** "***resulting in a RMSE of 833 ug/m3, 1.36 and 1.53***" is now changed to "***resulting in a PM$_{10}$ RMSE of 833 µg/m3, and AOD RMSE of 1.36 and 1.53***" in page 17, line 5-6.

*Page 15, Lines 11-12: I think "the observation-minus-simulation mismatch" means mean errors (ME). If the authors mean RMSE, this phrase has to be rewritten.*

**Reply:** To make it clear, "***and the observation-minus-simulation mismatch is reduced to 723 ug/m3, 1.30 and 1.33.***" is now changed to "***and the PM$_{10}$ RMSE is reduced to 743 µg/m3, and the AOD RMSE declined to 1.30 and 1.34, simultaneously***" in page 17, line 6.

*Page 15, Line 33: Fig. 4(b.1)   Is this a mistake for Fig. 4(a.1)?*

**Reply:** corrected.

*Page 16, Lines 26-28: It's ok with "only 160 tons against 49k tons" and "only 15k tons against 97k tons," but "only 74k tons against 50k tons" is not acceptable.*

**Reply:** The whole paragraph concluding the deposition in NCP and FWP is now modified in page 20, line 12-17 "***The total deposition in the NCP and FWP regions has been calculated and is shown in Figure 8(d). For the NCP region, 81k, 118k, and 70k tons of Mongolian dust were deposited during SD1, SD2, and SD3 events, while the total deposition from Chinese desert was about 8.3k, 20k, and 93k tons, respectively. For the other important cluster FWP, 4.3k, 22k, and 24k tons were attributed to domestic sources, and 20k, 57k, and 7.5k tons of dust were attributed to transnational transport from Mongolia. In general, Mongolian Gobi pose more severe threat to FWP and NCP region than the Chinese Gobi. About 63% of the dust deposition in FWP is attributed to the transnational transport. Over NCP, this value further grows up to 69%.***"

*Page 16, Line 31: 58% is attributed to the transnational transport over FWP, right? Please describe it in the text.*

**Reply**: See reply to the previous question.

*Page 18, Line 3: the authors stated in Conclusion that three super dust storms resulted in profoundly effects to Earth system, but the impact of the dust aerosols on weather, climate change, or the Earth System was not investigated in this study. If someone has already researched the impact of the dust storms in 2021 on the Earth System, those references have to be cited.*

**Reply:** We did not refer any paper about how the 2021 dust storm influence the Earth system exactly. Therefore, the statement is not solid, and is now changed from "***In spring 2021, three super dust storms occurred in East Asia after being absent for a (two) decade(s), which brought enormous health damages and property losses, and resulted in profoundly effects to Earth system.***" to "***In spring 2021, three super dust storms occurred in East Asia after being absent for one and a half decades, which brought enormous health damages and property losses.***" in page 20, line 19-20.

*Page 18, Line 29: I accessed http://106.37.208.233:20035, but couldn't obtain the PM10 data. I think Chinese government usually prohibits foreigners from accessing and obtaining Chinese environmental observation data. Could the authors provide the PM10 data used in this study based on the EGU data availability policy?*

**Reply**: The website can be only accessed through IE explorer. Remarks is added in page 22, line 14-16 by saying "***The real-time PM10 data are from the network established by the China Ministry of Environmental Protection and available to the public via http://106.37.208.233:20035/ (Internet Explorer 10 or 11), the PM10 data used in this paper can also be found in the Supplement.***"

*Technical corrections:*
*The authors often use "a (two) decade(s)" to mention the period between the 2000's and 2021 probably meaning one and half decades. However, this expression seems confusing and peculiar.*

**Reply:** "***a (two) decaded(s)***" is now changed to "***one and a half decades***".

*Besides, this manuscript contains too many errata and grammatical inaccuracies to publish as it is. I strongly recommend that the authors polish the manuscript more earnestly or use an English proofreading service.*
*For instance,*
*Page 2, Line 7: Fig.1(a) → Fig. 1(a) [Not only here, almost all of "Fig. xx" in this manuscript don't have a space after a period.]*

**Reply**: The mistakes are corrected throughout the paper.

*Page 2, Line 31: "BSCDREAM8b" → "BSC-DREAM8b"*

**Reply**: Accepted.

*Page 3, Line 8: "next the simulation models" What's this phrase?*

**Reply**: "***next the simulation models***" is changed to "***next to the simulation models***"

*Page 3, Line 31: Fig. 1)(a)*

**Reply**: Corrected.

*Page 4, Line 13: non-dust biase*

**Reply**: "***a non-dust biase***" is changed to "***non-dust bias***"

*Page 4, Line 31: "studied" → "was studied"???*

**Reply**: Accepted.

*Table 1: China Stand Time → Chinese Standard Time*

**Reply**: Accepted.

*Page 5, Line 11: Fig.1(b) → Figure 1(b) [Not only here, a word at the beginning of a sentence shouldn't be shortened.]*

**Reply**: "Fig. x" is changed to "Figure x" throughout the manuscript.

*Page 5, Line 15: Table.1 → Table 1*

**Reply**: Corrected.

*Page 5, Line 16: "2 to 3 day" → "2 to 3 days"???*

**Reply**: Corrected.

*Page 7, Line 29: Angstrom → The Angstrom exponent*

**Reply**: Corrected.

*Page 8, Line 1: Fig.5 → Figure 5*

**Reply**: Corrected.

*Page 8, Line 29: "but not far away to FWP or NCP" Is this really NOT far?*

**Reply**: It is a grammatical mistake. "***but not far away to FWP or NCP***" is now changed to "***but not as far as FWP or NCP***".

*Page 11, Line 7: by (Zender, 2003) → by Zender (2003)*

**Reply**: Corrected.

*Page 11, Line 11: What's F_h? Is it f_h?*

**Reply**: "$F_h$" is changed to "$f_h$".

*Page 11, Line 14: this sentence is wordy, colloquial, and extremely hard to read.*

**Reply**: "***The friction velocity threshold controls if dust is released from a surface at all, and if it is, how strong the emission is.***" is now changed to "***The friction velocity threshold controls if dust is released from a surface at all, and how strong the emission is***."

*Page 11 Line 22: in (Jin et al. 2018) → in Jin et al. (2018)*

**Reply**: Corrected.

*Page 12 Line 18: in (Jin et al. 2018) → in Jin et al. (2018)*

**Reply**: Corrected.

*Page 13, Lines 3-4: "by assimilation the bias-corrected …" What's this?*

**Reply**: "***by assimilation the bias-corrected …***" is now changed to "***by assimilating the bias-corrected …***".

*Page 13, Line 8: "in SD1 to SD3" → "in SD1, SD2, and SD3"*

**Reply**: Accepted.

*Page 13, Line 10: "Mongolia desert" → "Mongolian desert"*

**Reply**: "***Mongolia desert***" is modified to "***Mongolian desert***" all over the manuscript.

*Page 13, Line 11: "a new the emission field" what's this phrase?*

**Reply**: "***a new the emission***" is changed to "***the emission***".

*Page 13, Line 12: "with more grid cells from which emission took place located in…" This phrase is hard to understand.*

Reply: "***with more grid cells from which emission took place located in…***" is now changed to "***that emission took place in more grid cells that are located in the Alxa desert and in the eastern Mongolia***" in page 14, line 26-27.

*Page 13, Line 26: "map show that" → "map shows that"*

**Reply**: Corrected.

*Page 15, Line 22: Fig.3 → Figure 3*

**Reply**: Accepted.

*Page 15, Line 24-25: the tenses of the two verbs disagree.*

**Reply**: Corrected.

*Page 16, Line 20: "in SD1 to SD3" → "in SD1, SD2, and SD3"*

**Reply**: Corrected.

*Page 16, Line 21: "panel (a.1)" belongs to which figure?*

**Reply**: "As shown in panel (a.1)" is now changed to "As shown in Figure 8(a.1)".

*Page 16, Line 22: "almost not effect" → "almost do not effect"???*

**Reply**: "***but these were transported westward and almost not effect the densely populated areas.***" is now changed to "***but these were mainly transported westward and only a little fraction of them moved to the densely populated areas.***" in page 20, line 8-10.

*Page 16, Line 27: "Mongolia dust" → "Mongolian dust"*

**Reply**: Corrected.

*Page 18, Line 12: "the they are …" → "they are …"*

**Reply**: Corrected.

*Page 18, Line 25: "Mongolia dust" → "Mongolian dust"*

**Reply**: Corrected.

*Besides these errata, verb tense/article/punctuation mistakes are often found.*

**Reply:** We have carefully checked the entire manuscript, made many small language and grammar corrections throughout the whole paper. (There are too many tiny changes, so we did not highlight them.)

---

## Author Comment (AC2)

**Response to Referee #2**: We would like to thank the referee for the careful review and suggestion, which helps us to further improve the quality of the manuscript.

Our response follows (*the reviewer's comments are in italics and blue*)

*General comments: The authors present the emission inversion of 2021 super sandstorms in East Asia. The dust emissions are optimized based on a four-dimensional variational (4DVar) data assimilation and observations including both satellite AOD and groud-based PM10 data. With the newly inversed dust emission field, source apportionment experiments are also carried out. The study quantifies the relative contribution of Mongolian and Chinese source to the dust deposition in the two north China mega-city regions (Fenwei and North China Plain). Generally speaking, the paper is well written and scientific sound. There are however some aspects that should be explained before it can be published. I have some questions for the authors and some comments that could help to improve the manuscript.*

*1)Page 11, line 5. They should acknowledge emission uncertainty might not be the dominant source in the three dust events. Afterall, they state in Section 4.1.2, page 15, line 14 "This suggests that the dust plume in the higher layers moves faster, and is in further southeast than the dust cloud at bottom layer. However, this feature is not correctly captured by our LOTOS-EUROS/dust model." They indicate here error might also rise from the advection transport next to the emission uncertainty. Then it is also recommended to discuss the difficult or possibility of handling the emission and transport error at the same time.*

**Reply:** Uncertainty in the transport is of course part of the dust simulation error source and should be emphasized. Remarks are now added in page 11, line 23-25 by saying "***In this study, we define the main model uncertainty to be in the parametrization of the dust emissions. Although other model processes such as transport and deposition are uncertain too, for the events studied here these assumed to be of less importance than the location and the intensity of dust emission.***"

Discussions are also added about how to handle the correct the error from the advection transport in page 17, line 13-17: "***The mismatch in simulated vertical structure is mostly likely caused by uncertainty in the advection transport. The two-dimensional grid distortion technique (Jin et al., 2021) which is independent on the emission inversion could adjust the horizontal position of the dust cloud simulation to better fit the available measurements, but***

***is not yet able to adjust the vertical structure as is required here. A three-dimensional grid distortion with data that measuring the vertical profile of dust cloud is planned to solve this issue in our future research.***"

*2)Page 16 Sect 4.2, the authors are omitting a discussion why source apportion is focusing on the Mongolia and China desert source. Is it because the different desertification situation in Mongolia or China, or their results could help the greenness recover in China and Mongolia in the future? This is important for reader while introducing the source apportionment experiments.*

**Reply**: Definitely, it should be explained that why we focus on the contribution of Mongolian dust and Chinese dust. Remarks are added in page 18, line 30-33 and page 19, line 1-5, by saying ***"China government has launched several large-scale ecological engineering projects to combat the environmental problems in the northern China during recent decades. One of the largest is the Three-North Shelter Forest Program, which aimed at increasing the vegetation cover upto 15% by 2050 (Niu et al., 2019). Several studies (Shao et al., 2013; Tan and Li, 2015) reported that the vegetation recover weakened dust storms substantially. In contrast, Mongolia has experienced the ever-increasing land degradation and desertification (Meng et al., 2020), which aggravates the spring dust storms (Han et al., 2021). To evaluate the roles of Mongolian and Chinese Gobi deserts in the 2021 super sandstorms quantitatively, source apportionment tests based on the estimated emission field are carried out further. These source apportionment tests focus on the two dust-affected mega-city clusters in the northern China, North China Plain (NCP) and Fenwei Plain (FWP), and aims to calculated whether the dust originates from transnational transport from Mongolia or from domestic sources in China."***

*Specific comments:*

*Page 1, Line 18: 'partical' → 'particle'*
 **Reply:** corrected.

*Page 2, Line 23: 'consists' → 'consist'*
 **Reply:** corrected.

*Page 2, Line 34: 'differencec' → 'differences'*

**Reply:** corrected.

*Page 3, Line 26: 'where' → 'were' ?*

**Reply:** corrected.

*Page 4, Line 29: 'Similar for PM10 a non-dust bias correction as used in Jin et al. (2019a, 2021) is adopted in this work' → 'Similar for PM10, a non-dust bias correction as used in Jin tt al. (2019a, 2021), is adopted in this work' .*

**Reply:** accepted.

*Page 11, Line 11: 'Fh' → 'fh'?*

**Reply:** accepted.

*Page 13, Line 11: remove 'the'*

**Reply:** "*a new the emission is estimated*" is now changed to "*the emission field is estimated*"

*Page 13, Line 31: 'emisison' → 'emission'.*

**Reply:** corrected.

*Page 15, Line 13: 'visilbe' → 'visible'.*

**Reply:** corrected.

*Page 15, Line 24: should be 'The standard model simulated AOD values are even larger than 4'.*

**Reply:** accepted.

*Page 15, Line 32: 'concentations' → 'concentrations'.*

**Reply:** accepted.

*Page 16, Line 7: 'accurate' → 'accurately'.*

**Reply:** accepted.

*Page 16, Line 22: 'effect' →'affecting'?*

**Reply:** corrected.

*Figure 6 the title of the panels using SDS should be consistent with the text using SD.*

**Reply**: The Figure 6 is updated as follow.

[Figure]

**Figure 6.** Distribution of the a priori (a.1-c.1) and posterior (a.2-c.2) accumulated dust emission for the SD1, SD2 and SD3; the total priori and posterior emission either from China or from Mongolia during SD1, SD2 and SD3 (d).